



# Seamless mapping of long-term (2010-2020) daily global XCO₂ and XCH₄ from GOSAT, OCO-2, and CAMS-EGG4 with a spatiotemporally self-supervised fusion method

Yuan Wang[1], Qiangqiang Yuan[1,2], Tongwen Li[3], Yuanjian Yang[4], Siqin Zhou[1], Liangpei Zhang[5]

[1]School of Geodesy and Geomatics, Wuhan University, Wuhan, Hubei, 430079, China.

[2]The Key Laboratory of Geospace Environment and Geodesy, Ministry of Education, Wuhan University, Wuhan, Hubei, 430079, China.

[3]School of Geospatial Engineering and Science, Sun Yat-sen University, Guangzhou, Guangdong, 519082, China.

[4]School of Atmospheric Physics, Nanjing University of Information Science & Technology, Nanjing, Jiangsu, 210044, China.

[5]The State Key Laboratory of Information Engineering in Surveying, Mapping and Remote Sensing, Wuhan University, Wuhan, Hubei, 430079, China.

*Correspondence to*: Qiangqiang Yuan (qyuan@sgg.whu.edu.cn)

**Abstract.** Precise and continuous monitoring on long-term carbon dioxide ($CO_2$) and methane ($CH_4$) over the globe is of great importance, which can help study global warming and achieve the goal of carbon neutrality. Nevertheless, the available observations of $CO_2$ and $CH_4$ from satellites are generally sparse, and current fusion methods to reconstruct their long-term values on a global scale are few. To address this problem, we propose a novel spatiotemporally self-supervised fusion method to establish long-term daily seamless XCO₂ and XCH₄ products from 2010 to 2020 over the globe at grids of 0.25°. A total of three datasets are applied in our study, including GOSAT, OCO-2, and CAMS-EGG4. Attributed to the significant sparsity of data from GOSAT and OCO-2, the spatiotemporal Discrete Cosine Transform is considered for our fusion task. Validation results show that the proposed method achieves a satisfactory accuracy, with the $\sigma$ ($R^2$) of ~ 1.18 ppm (> 0.9) and 11.3 ppb (0.9) for XCO₂ and XCH₄ against TCCON measurements, respectively. Overall, the performance of fused results distinctly exceeds that of CAMS-EGG4, which is also superior or close to those of GOSAT and OCO-2. Especially, our fusion method can effectively correct the large biases in CAMS-EGG4 due to the issues from assimilation data, such as the unadjusted anthropogenic emission inventories for COVID-19 lockdowns in 2020. Moreover, the fused results present coincident spatial patterns with GOSAT and OCO-2, which accurately display the long-term and seasonal changes of globally distributed XCO₂ and XCH₄. The daily global seamless gridded (0.25°) XCO₂ and XCH₄ from 2010 to 2020 can be freely accessed at http://doi.org/10.5281/zenodo.7388893 (Wang et al., 2022b).



## 1 Introduction

As the most abundant greenhouse gases (GHGs) due to human activities, atmospheric carbon dioxide ($CO_2$) and methane ($CH_4$) play significant roles in climate change and directly contribute to global warming (Meinshausen et al., 2009; Montzka et al., 2011; Solomon et al., 2010; Yoro and Daramola, 2020; Shine et al., 2005). For decades, the rising anthropogenic surface emissions of $CO_2$ and $CH_4$ result in their long-term rapid uptrends (Choulga et al., 2021; Moran et al., 2022; Lin et al., 2021; Petrescu et al., 2021), which have greatly affected the carbon cycle (Battin et al., 2009; Sjögersten et al., 2014) and ecosystem balance (Liu and Greaver, 2009; Hotchkiss et al., 2015). According to measurements from the Global Greenhouse Gas Reference Network (https://gml.noaa.gov/ccgg/), annual surface $CO_2$ and $CH_4$ mole fractions break 412 parts per million (ppm) and 1878 parts per billion (ppb) in 2020, with growths of ~ 68 ppm and 222 ppb since 1985, respectively. To mitigate global warming, the Paris Agreement (https://unfccc.int/process-and-meetings/the-paris-agreement/) has indicated that the increment of temperature should not exceed 2 °C (preferably to 1.5 °C) by comparison with the pre-industrial level. This requires all efforts from the whole society to reach the global peaking of GHGs surface emissions as early as possible, especially for $CO_2$ and $CH_4$, which eventually create a carbon-neutral world by mid-century. Therefore, it is an urgent need to precisely and continuously monitor atmospheric $CO_2$ and $CH_4$ on a global scale.

To date, remote sensing observations have been extensively adopted in plenty of domains (Wang et al., 2021a, 2022c; Zhou et al., 2022), which also emerged as regular techniques to acquire globe-scale atmospheric $CO_2$ and $CH_4$ spatial patterns (He et al., 2022a; Buchwitz et al., 2015; Bergamaschi et al., 2013). For instance, the EnviSat can provide global column-mean dry-air mole fraction of $CO_2$ ($XCO_2$) and $CH_4$ ($XCH_4$) at a coarse resolution of $30 \times 60$ km$^2$, with the payload of the Scanning Imaging Absorption Spectrometer for Atmospheric Cartography (Burrows et al., 1995; Beirle et al., 2018). The Thermal and Near-Infrared Sensor for carbon Observations - Fourier Transform Spectrometer onboard the Greenhouse Gases Observing Satellite (GOSAT) (Hamazaki et al., 2005; Velazco et al., 2019) can produce ~ 10-km $XCO_2$ and $XCH_4$ over the globe based on three spectral bands. The Orbiting Carbon Observatory 2/3 (OCO-2/3) (Crisp et al., 2017; Doughty et al., 2022) carries three-channel grating spectrometers to generate globally covered $XCO_2$ at a much finer spatial resolution of $1.29 \times 2.25$ km$^2$. The Carbon Dioxide Spectrometer named CarbonSpec onboard the TanSat (Liu et al., 2018) of China launched in 2016, which can accurately map high-resolution (~ 2 km) global $XCO_2$ spatial distribution.

As for long-term observations of $XCO_2$ and $XCH_4$, the operational products from GOSAT and OCO-2 are widely applied in carbon-related applications, such as the computation of carbon fluxes (Fraser et al., 2013; Wang et al., 2019), inferring carbon sources and sinks (Deng et al., 2014; Houweling et al., 2015), quantifying $CO_2$ and $CH_4$ emissions (Turner et al., 2015; Hakkarainen et al., 2016), and estimation of terrestrial net ecosystem exchange (Jiang et al., 2022). Nevertheless, large-scale missing data consists in the $XCO_2$ and $XCH_4$ products from GOSAT and OCO-2, which is attributed to the narrow swath of their observations (Crisp et al., 2017) and contamination of cloud and aerosol (Taylor et al., 2016). Seamless information of



$XCO_2$ and $XCH_4$ can help better understand the driving factors of long-term variations for $CO_2$ and $CH_4$ due to surface
emissions and atmospheric transport (Kenea et al., 2023; Liu et al., 2020). In addition, full-coverage $XCO_2$ and $XCH_4$ products
are more useful to analyze carbon source-sink dynamics (Reithmaier et al., 2021; Crosswell et al., 2017) and impacts on climate
changes caused by the elevated $CO_2$ and $CH_4$ (Chen et al., 2021; Le Quéré et al., 2019). Hence, it is significant and essential
to assure the spatiotemporal continuity of $XCO_2$ and $XCH_4$ products from GOSAT and OCO-2, which is conducive to achieving
the goal of carbon neutrality.
A lot of efforts have been made to generate seamless $XCO_2$ and $XCH_4$ products for GOSAT and OCO-2. Initially, interpolation-
based methods are widely utilized, such as the fixed rank kriging interpolation (Katzfuss and Cressie, 2011), semantic kriging
interpolation (Bhattacharjee et al., 2014), and space-time kriging interpolation (He et al., 2020; Li et al., 2022). However, the
interpolated results are usually performed at coarse spatial resolutions (e.g., 1°) and tend to show high uncertainties and over-
smoothed distribution due to the extreme sparsity of original data. At present, data fusion techniques (He et al., 2022a, b; Zhang
et al., 2022; Zhang and Liu, 2023; Siabi et al., 2019) have emerged as new methods to acquire full-coverage products for
GOSAT and OCO-2 at a high spatial resolution, which absorb advantages from multisource data. Generally, these methods
exploited machine learning algorithms to train an end-to-end fusion function with multiple seamless data (e.g., model and
reanalysis) as inputs. For example, Siabi et al. (2019) employed multi-layer perceptron and eight environmental variables (e.g.,
net primary productivity and leaf area index) to map full-coverage $XCO_2$ in Iran; He et al. (2022b) established seamless results
over China using the OCO-2 $XCO_2$ product, CarbonTracker model data, and auxiliary co-variates based on the light gradient
boosting machine; Zhang et al. (2022) proposed a geographically weighted neural network to produce full-coverage $XCO_2$
product across China by fusing the datasets from OCO-2, CAMS-EGG4 (reanalysis), and ERA5; and Zhang and Liu (2023)
adopted multiple datasets, e.g., EnviSat, GOSAT, OCO-2, CarbonTracker, and ERA5, and obtained long-term seamless $XCO_2$
product in China through a finely devised neural network.
These data fusion approaches provided high-quality results with seamless distribution and greatly enhance the data availability
for GOSAT and OCO-2. Nevertheless, the application areas of current fused products merely target at local or national scales,
which are insufficient for globe-scale researches. Meanwhile, existing data fusion frameworks are regarded as end-to-end
functions, which lack consideration for spatiotemporal self-correlation of original data (e.g., OCO-2). They normally require
massive auxiliary co-variates (e.g., ERA5) as inputs and consume a large time in training procedures. Moreover, only $XCO_2$
products are taken into account while the data fusion studies for $XCH_4$ products are scarce. In conclusion, it is valuable and
imperative to generate long-term globally distributed seamless $XCO_2$ and $XCH_4$ products for GOSAT and OCO-2 with an
efficient data fusion method, which considers the knowledge of their spatiotemporal self-correlation.
The present study focuses on generating long-term daily global seamless $XCO_2$ and $XCH_4$ products from 2010 to 2020 at the
grids of 0.25° via a spatiotemporally self-supervised fusion method. A total of three datasets are utilized in our study without



any auxiliary co-variates, including GOSAT, OCO-2, and CAMS-EGG4. CAMS-EGG4 can provide long-term gridded full-coverage $XCO_2$ and $XCH_4$ datasets over the globe, which is suitable for our fusion task. Since the data from GOSAT and OCO-2 is significantly sparse in space-time domain (see Fig. 1), the fusion procedures are difficult to be performed. By contrast, frequency domain contains comprehensive information due to its more concentrated signal distribution. Discrete Cosine Transform (DCT) (Rao and Yip, 2014) is an efficient algorithm to convert signal into frequency domain. In this study, a novel self-supervised fusion method based on spatiotemporal DCT (S-STDCT) is developed for the fusion task. Details of the S-STDCT fusion method are presented in Section 3. Validation results show that the S-STDCT fusion method achieves a satisfactory performance. Generally, the accuracy of fused results largely exceeds that of CAMS-EGG4, which is also better than or close to those of GSOAT and OCO-2.

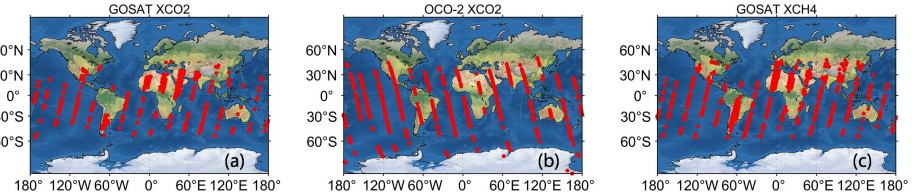

**Figure 1.** An example of daily spatial footprints for (a) GOSAT $XCO_2$, (b) OCO-2 $XCO_2$, and (c) GOSAT $XCH_4$. Red points signify the available data. Background maps are naturally shaded reliefs over the globe.

This paper arranges the remaining sections as follows. Section 2 describes the data records employed in our study, including the $XCO_2$ and $XCH_4$ from in-situ stations, GOSAT, and CAMS-EGG4 and $XCO_2$ from OCO-2. Section 3 provides the specification of the developed S-STDCT fusion method. Section 4 presents the experiment results, which consist of elaborative validations against in-situ measurements and assessments of spatial distribution on multi-temporal scales. At last, conclusions and future works are summarized in section 5.

## 2 Data description

### 2.1 GOSAT $XCO_2$ and $XCH_4$ products

A famous $XCO_2$ retrieval algorithm devised for GOSAT (Taylor et al., 2022), i.e., the Atmospheric $CO_2$ Observations from Space (ACOS), employs three infrared spectral bands at ~ 0.76, 1.6, and 2.0 μm, which are denoted as Oxygen-A, $CO_2$ weak, and $CO_2$ strong, respectively. Regarding $XCH_4$, the latest retrieval algorithm for GOSAT from the University of Leicester is recently updated, which considers the ratio of $XCH_4$:$XCO_2$ as a proxy (Parker et al., 2020). It is based on the theory that the impacts from atmospheric scattering and sensor are mostly similar for $XCH_4$ and $XCO_2$ in a shared absorption band at ~ 1.6 μm. The GOSAT $XCO_2$ and $XCH_4$ products are both performed at spatial resolutions of 10.5 km (diameter) over the globe with revisit times of 3 days. In our study, the scientific data records of "XCO2" in ACOS_L2_Lite_FP (level 2, bias-corrected, V9r) and "XCH4" in UoL-GHG-L2-CH4-GOSAT-OCPR (level 2, V9) are adopted. Furthermore, the QA records of "XCO2



Quality Flag" and "XCH4 Quality Flag" are exploited to filter bad data. Relevant information of $XCO_2$ and $XCH_4$ products
from GOSAT is shown in Table 1.
**Table 1.** Detailed information of the datasets considered in this study.

| Source | Scientific data record | Version | Spatial resolution | Temporal resolution | Period |
|---|---|---|---|---|---|
| GOSAT | XCO2<br>XCO2 Quality Flag | V9r | 10.5 km (diameter) | Daily (~ 13:00 local time) | 2010-2014 |
| | XCH4<br>XCH4 Quality Flag | V9 | | | 2010-2020 |
| OCO-2 | XCO2<br>XCO2 Quality Flag | V10r | 1.29×2.25 km² | Daily (~ 13:36 local time) | 2015-2017 |
| | XCO2<br>XCO2 Quality Flag | V11r | | | 2018-2020 |
| CAMS-EGG4 | CO2 column-mean molar fraction<br>CH4 column-mean molar fraction | - | 0.75° | 3 hours | 2010-2020 |

**2.2 OCO-2 XCO₂ product**
Apart from GOSAT, the ACOS $XCO_2$ retrieval algorithm is also applied to OCO-2 observations (Kiel et al., 2019), which
utilizes the same bands of the Oxygen-A, $CO_2$ weak, and $CO_2$ strong. OCO-2 provides a global $XCO_2$ product at a high spatial
resolution of 1.29×2.25 km² with a revisit time of 16 days. After 2015, the $XCO_2$ product from OCO-2 is used for fusion
instead of GOSAT due to its more observation counts and better accuracy. In this study, the scientific data record of "XCO2"
in OCO2_L2_Lite_FP (level 2, bias-corrected) is applied in the fusion with CAMS-EGG4 using the developed method.
Moreover, the quality assurance (QA) record of "XCO2 Quality Flag" is adopted to filter bad data. Since the OCO-2 $XCO_2$
product of the latest version (V11r) is still on processing, both data of V10r and V11r are considered in our study. Related
information of $XCO_2$ product from OCO-2 is given in Table 1.
**2.3 CAMS-EGG4 GHGs reanalysis datasets**
CAMS-EGG4 is recent globally distributed operational GHGs reanalysis datasets supported by the European Centre for
Medium-range Weather Forecasts (Agusti-Panareda et al., 2022). It assimilates the forecasts from the Integrated Forecasting
System with multiple satellite products, which include Envisat, GOSAT, and Metop-A/B (August et al., 2012), via physical
and chemistry principles. The CAMS-EGG4 can generate long-term gridded seamless $XCO_2$ and $XCH_4$ datasets and related
fields at spatial and temporal resolutions of 0.75º and 3 hours, respectively. Unfortunately, there are a few limitations in CAMS-
EGG4, such as the uncorrected anthropogenic emissions for COronaVIrus Disease 2019 (COVID-19) lockdowns, which are
scheduled to be fixed by the official team in the future (Agusti-Panareda et al., 2022). It is worth noting that the $XCO_2$ and
$XCH_4$ products from GOSAT and OCO-2 employed in this paper are not assimilated in CAMS-EGG4. In our study, the
scientific data records of "CO2 column-mean molar fraction" and "CH4 column-mean molar fraction" are exploited for the
fusion with GOSAT and OCO-2 through the developed method. Details of CAMS-EGG4 datasets are provided in Table 1.





**2.4 TCCON measurements**
In our study, the $XCO_2$ and $XCH_4$ measurements provided by an international in-situ network, which is named after TCCON
(Wunch et al., 2011) (https://tccondata.org/), are utilized to validate the fused results. The in-situ measurements of TCCON
are extensively used in the validation for $XCO_2$ and $XCH_4$ products from GOSAT, OCO-2, and CAMS-EGG4 (Hong et al.,
2022; Yoshida et al., 2013; Wunch et al., 2017; Wu et al., 2018; Agusti-Panareda et al., 2022). Figure 2 depicts the spatial
locations of TCCON stations, with the marks of white-edged red circles. The measurements of version GGG2020 (Laughner
et al., 2022) from 29 stations around the world are adopted. Specific information of the stations is listed in Table 2.

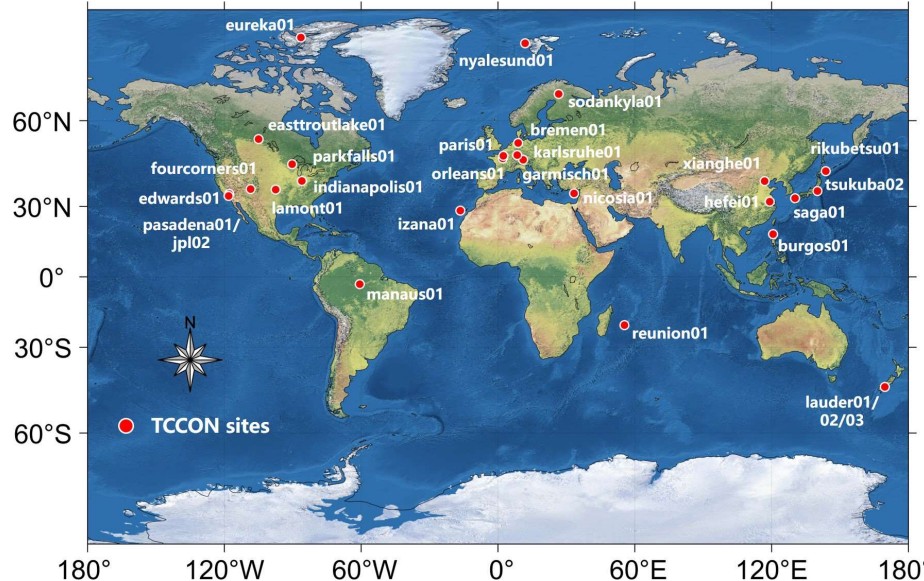


**Figure 2.** Spatial locations of in-situ stations from TCCON used in the present study. The background map is a naturally shaded relief over
the globe.
**3 Methodology**
**3.1 Data pre-processing**
Data pre-processing is an important procedure to ensure the rationality and reliability of fused results. In this study, the values
of "QA=0" in $XCO_2$ and $XCH_4$ from GOSAT and OCO-2 are discarded, which filters the bad data. Besides, the CAMS-EGG4
$XCO_2$ and $XCH_4$ at a temporal resolution of 3 hours are averaged in a single day to produce daily datasets. Finally, the spatial
resolutions of $XCO_2$ and $XCH_4$ from GOSAT, OCO-2, and CAMS-EGG4 ought to be adjusted to the same value. A globally
covered grid of 721×1441 (0.25º) is employed in our study. The $XCO_2$ and $XCH_4$ from GOSAT, OCO-2, and CAMS-EGG4
are re-gridded to 0.25° using the area-weighted aggregation (Wang et al., 2021b) and Inverse Distance Weighted (Mueller et
al., 2004) interpolation, respectively.



**Table 2.** Detailed information of TCCON in-situ stations adopted in our study. No.: number.

| No. | Site name | Latitude | Longitude | Location | Start date | End date |
|---|---|---|---|---|---|---|
| 1 | bremen01 | 53.10 | 8.85 | Europe | 2010-01-01 | 2020-12-31 |
| 2 | burgos01 | 18.53 | 120.65 | Asia | 2017-03-03 | 2020-04-30 |
| 3 | easttroutlake01 | 54.36 | -104.99 | North America | 2016-10-03 | 2020-12-31 |
| 4 | edwards01 | 34.96 | -117.88 | North America | 2013-07-20 | 2020-12-31 |
| 5 | eureka01 | 80.05 | -86.42 | North America | 2010-07-24 | 2020-07-07 |
| 6 | fourcorners01 | 36.80 | -108.48 | North America | 2013-03-16 | 2013-10-03 |
| 7 | garmisch01 | 47.48 | 11.06 | Europe | 2010-01-01 | 2020-12-31 |
| 8 | hefei01 | 31.90 | 119.17 | Asia | 2016-01-08 | 2020-12-31 |
| 9 | indianapolis01 | 39.86 | -86.00 | North America | 2012-08-23 | 2012-12-01 |
| 10 | izana01 | 28.31 | -16.50 | Atlantic Ocean | 2014-01-02 | 2020-12-31 |
| 11 | jpl02 | 34.20 | -118.18 | North America | 2011-05-19 | 2018-05-14 |
| 12 | karlsruhe01 | 49.10 | 8.44 | Europe | 2014-01-15 | 2020-12-31 |
| 13 | lauder01 | 36.60 | -97.49 | Oceania | 2010-01-01 | 2010-02-19 |
| 14 | lauder02 | -45.04 | 169.68 | Oceania | 2013-01-02 | 2018-09-30 |
| 15 | lauder03 | -45.04 | 169.68 | Oceania | 2018-10-02 | 2020-12-31 |
| 16 | lamont01 | -45.04 | 169.68 | North America | 2010-01-01 | 2020-12-31 |
| 17 | manaus01 | -3.21 | -60.60 | South America | 2014-09-30 | 2015-07-27 |
| 18 | nicosia01 | 35.14 | 33.38 | Asia | 2019-09-03 | 2020-12-31 |
| 19 | nyalesund01 | 78.92 | 11.92 | Arctic Ocean | 2010-01-01 | 2020-12-31 |
| 20 | orleans01 | 47.96 | 2.11 | Europe | 2010-01-01 | 2020-12-31 |
| 21 | paris01 | 48.85 | 2.36 | Europe | 2014-09-23 | 2020-12-31 |
| 22 | parkfalls01 | 45.94 | -90.27 | North America | 2010-01-01 | 2020-12-31 |
| 23 | pasadena01 | 34.14 | -118.13 | North America | 2012-09-20 | 2020-12-31 |
| 24 | reunion01 | -20.90 | 55.48 | Indian Ocean | 2015-03-01 | 2020-07-18 |
| 25 | rikubetsu01 | 43.46 | 143.77 | Asia | 2014-06-24 | 2020-12-31 |
| 26 | saga01 | 33.24 | 130.29 | Asia | 2011-07-28 | 2020-12-31 |
| 27 | sodankyla01 | 67.37 | 26.63 | Europe | 2018-03-05 | 2020-12-31 |
| 28 | tsukuba02 | 36.05 | 140.12 | Asia | 2014-03-28 | 2020-12-31 |
| 29 | xianghe01 | 39.80 | 116.96 | Asia | 2018-06-14 | 2020-12-31 |

## 3.2 Spatiotemporally self-supervised fusion method

Since the sparsity of data from GOSAT and OCO-2 is significant in space-time domain (see Fig. 1), it is difficult to perform

fusion procedures for them. In contrast, frequency domain is more suitable because of its concentrated signal distribution. DCT

is an efficient algorithm to transform signal into frequency domain (Rao and Yip, 2014), which has been widely applied in

image compression (Cintra and Bayer, 2011), geophysical data filtering (El-Mahallawy and Hashim, 2013), and remote sensing

data reconstruction (Wang et al., 2012, 2022a; Fredj et al., 2016; Pham et al., 2019). In our study, a novel self-supervised fusion

method based on spatiotemporal DCT, i.e., S-STDCT, is developed for the fusion task, which fully adopts the spatiotemporal

knowledge of self-correlation in GOSAT and OCO-2 products.

### 3.2.1 Spatiotemporal DCT

A total of eight types of DCT are proposed, among which the second type (type-II) is commonly utilized due to its simple

calculation and broad application range (Rao and Yip, 2014). Hence, the type-II DCT is considered in this study. The

spatiotemporal DCT is a 3-dimensional form (hereafter *STDCT*), which can be expressed as Eq. (1):



$$X(u,v,w) = c(u)c(v)c(w) \sum_{i=0}^{M-1} \sum_{j=0}^{N-1} \sum_{t=0}^{P-1} x(i,j,t) cos\left[\frac{(i+0.5)\pi}{M}u\right] cos\left[\frac{(j+0.5)\pi}{N}v\right] cos\left[\frac{(t+0.5)\pi}{P}w\right],$$ (1)
where $c(u) = \begin{cases} \sqrt{\frac{1}{M}}, u=0 \\ \sqrt{\frac{2}{M}}, u\neq0 \end{cases}$, $c(v) = \begin{cases} \sqrt{\frac{1}{N}}, v=0 \\ \sqrt{\frac{2}{N}}, v\neq0 \end{cases}$, $c(w) = \begin{cases} \sqrt{\frac{1}{P}}, w=0 \\ \sqrt{\frac{2}{P}}, w\neq0 \end{cases}$; $x$ indicates the original 3-dimensional tensor; $i$, $j$,
and $t$ represent the row, column, and temporal sequence, respectively ($i \in$ [0, M-1], $j \in$ [0, N-1], and $t \in$ [0, P-1]); $X$
signifies the transformed 3-dimensional tensor; $u$, $v$, and $w$ denote the transformed coordinates in frequency domain, which
share the same ranges with $i$, $j$, and $t$ (e.g., $u \in$ [0, M-1]), respectively. The inverse transformation of $STDCT$ (hereafter
$ISTDCT$) is provided in Eq. (2):
$$x(i,j,t) = c(u)c(v)c(w) \sum_{u=0}^{M-1} \sum_{v=0}^{N-1} \sum_{w=0}^{P-1} X(u,v,w) cos\left[\frac{(i+0.5)\pi}{M}u\right] cos\left[\frac{(j+0.5)\pi}{N}v\right] cos\left[\frac{(t+0.5)\pi}{P}w\right],$$ (2)

**180 3.2.2 Self-supervised fusion scheme with spatiotemporal knowledge**

It has been documented that the $XCO_2$ and $XCH_4$ products derived from remote sensing satellites generally present better
accuracy compared to reanalysis datasets (Agusti-Panareda et al., 2022; He et al., 2022a; Parker et al., 2020). Therefore, the
brand new $XCO_2$ and $XCH_4$ products from GOSAT and OCO-2 are regarded as the criteria (or ground truths), which will be
fused with CAMS-EGG4 datasets. At first, a spatially and temporally varying function relationship (see Eq. (3)) is
hypothesized between GOSAT/OCO-2 and CAMS-EGG4 $XCO_2$/$XCH_4$ values.
$XG_s = f(XGc, Row, Col, Time)$, (3)
where $XG_s$ denotes the $XCO_2$/$XCH_4$ values from GOSAT/OCO-2; $XG_c$ indicates the $XCO_2$/$XCH_4$ values from CAMS-EGG4;
$Row$, $Col$, and $Time$ represent the row (or latitude), column (or longitude), and temporal sequence, respectively. To conveniently
solve this problem, Eq. (3) is simplified into the scalar product form of $XG_c$ and a spatially and temporally varying tensor
(defined as $\delta$), as shown in Eq. (4):
$XG_s = XGc * \delta(Row, Col, Time)$, (4)
Afterward, the factor (i.e., $\delta$) can be acquired using the $XCO_2$/$XCH_4$ values at the grids where the GOSAT/OCO-2 and CAMS-
EGG4 data are both available. In our study, a self-supervised fusion scheme is introduced to solve Eq. (4) based on the
spatiotemporal knowledge of self-correlation in GOSAT and OCO-2 products. Due to the large sparsity of data from GOSAT
and OCO-2 in space-time domain, the $STDCT$ is applied for the fusion task.
Inspired by previous studies adopting the $STDCT$ (Garcia, 2010; Wang et al., 2012, 2022a; Fredj et al., 2016; Pham et al.,
2019), the S-STDCT fusion method searches for the spatially and temporally varying tensor, i.e., $\delta$, that minimizes Eq. (5),
including a residual (left) and a smoothing (right) term.
$$E(\delta) = \left\| \varphi^{\frac{1}{2}} * (\hat{\delta} - \delta) \right\|^2 + \varepsilon \left\| \nabla^2 \delta \right\|^2,$$ (5)



where $\| \ \|$ signifies the Euclidean norm; $\varphi$ represents the binary mask showing the data is whether available or not; $\varepsilon$ and
$\nabla^2$ indicate a smoothing factor and the Laplace operator, respectively. This equation can be solved by iterations via Eq. (6):
$\hat{\delta} = \gamma ISTDCT\left(\rho * STDCT\left(\varphi * \left(\delta - \hat{\delta}\right) + \hat{\delta}\right)\right) + (1 - \gamma)\hat{\delta},$                (6)
where $\gamma$ is a relaxation factor to accelerate convergence; $\rho$ indicates a 3-dimensional filter related to the smoothing term,
which is defined in Eq. (7):
$\rho(d_1, d_2, d_3) = \dfrac{1}{1 + \varepsilon \sum_{k=1}^{3} 2\left[1 - \cos\dfrac{(d_k - 1)\pi}{n_k}\right]},$                (7)
Here, $d_k$ represents the $d^{th}$ value along the $k^{th}$ dimension ($k$ = 1, 2, and 3); $n_k$ denotes the size of $\delta$ along the $k^{th}$ dimension.
Namely, $d_1$, $d_2$, and $d_3$ stand for $u$, $v$, and $w$ (see Eq. (1)), respectively. In this study, the number of total iterations, $\gamma$, and
$\varepsilon$ are empirically configured to 100, 1.5, and a range from $10^3$ to $10^{-1}$ (spaced with 100 intervals), respectively. It is worth
noting that $\hat{\delta}$ is initialized through the temporal nearest neighbor interpolation. Regarding the grids where the data is missing
during the whole temporal sequence, $\hat{\delta}$ is initially set to 1. More details about the solution steps can be found in Garcia (2010).

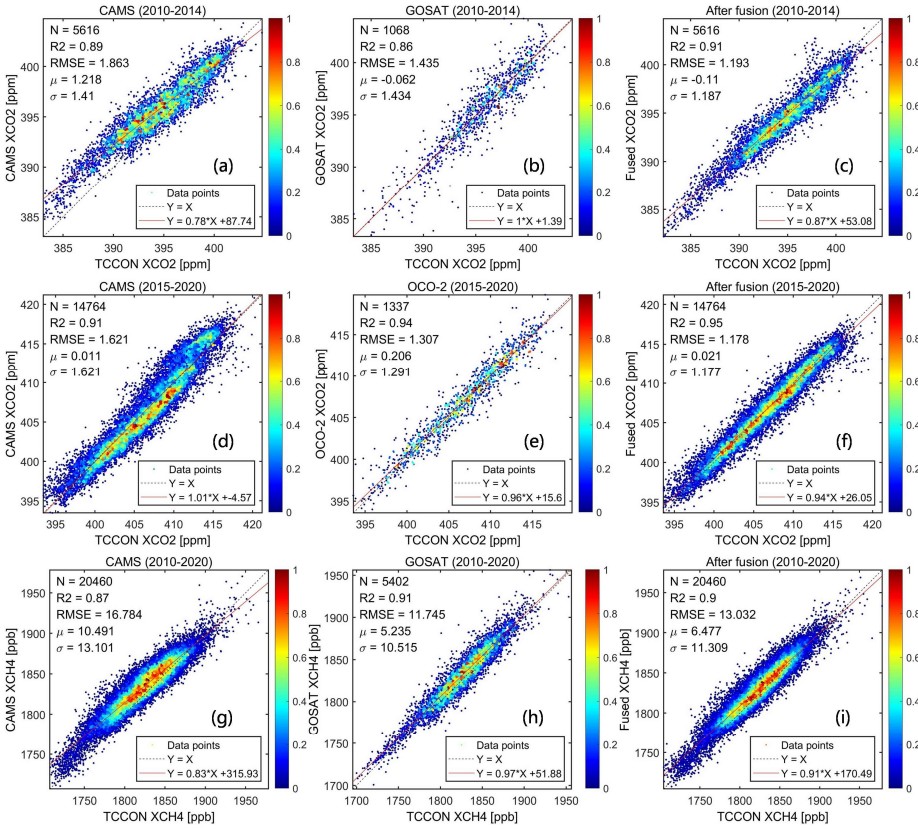


**Figure 3.** Density scatter-plots of the in-situ validation results for (a, d, and g) CAMS-EGG4, (b and h) GOSAT, (e) OCO-2, and (c, f, and
i) fused results. Black dotted and red full lines stand for the 1:1 and fitted lines, respectively. Color ramps show the normalized densities of
data points. X: TCCON data; Y: CAMS-EGG4/GOSAT/OCO-2/fused data. Unit: ppm/ppb to $XCO_2$/$XCH_4$ for RMSE, $\mu$, and $\sigma$.



### 3.3 Evaluation schemes

In our study, the evaluation schemes include in-situ validations and assessments of spatial distribution. To be specific, the GOSAT, OCO-2, CAMS-EGG4, and fused $XCO_2$ and $XCH_4$ are validated against TCCON measurements, which consists of the comparisons for overall and individual in-situ stations. The spatial distribution of the GOSAT, OCO-2, CAMS-EGG4, and fused $XCO_2$ and $XCH_4$ are assessed on multi-temporal scales, i.e., multi-year mean, seasonal, and annual. A total of four metrics are exploited, covering the Determination-Coefficient ($R^2$), Root-Mean-Square-Error (RMSE), Mean-Bias ($\mu$), and Standard-Deviation of Bias ($\sigma$). The significance levels of $p < 0.01$ are applied in the computations of all metrics.

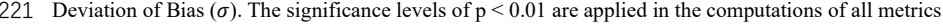

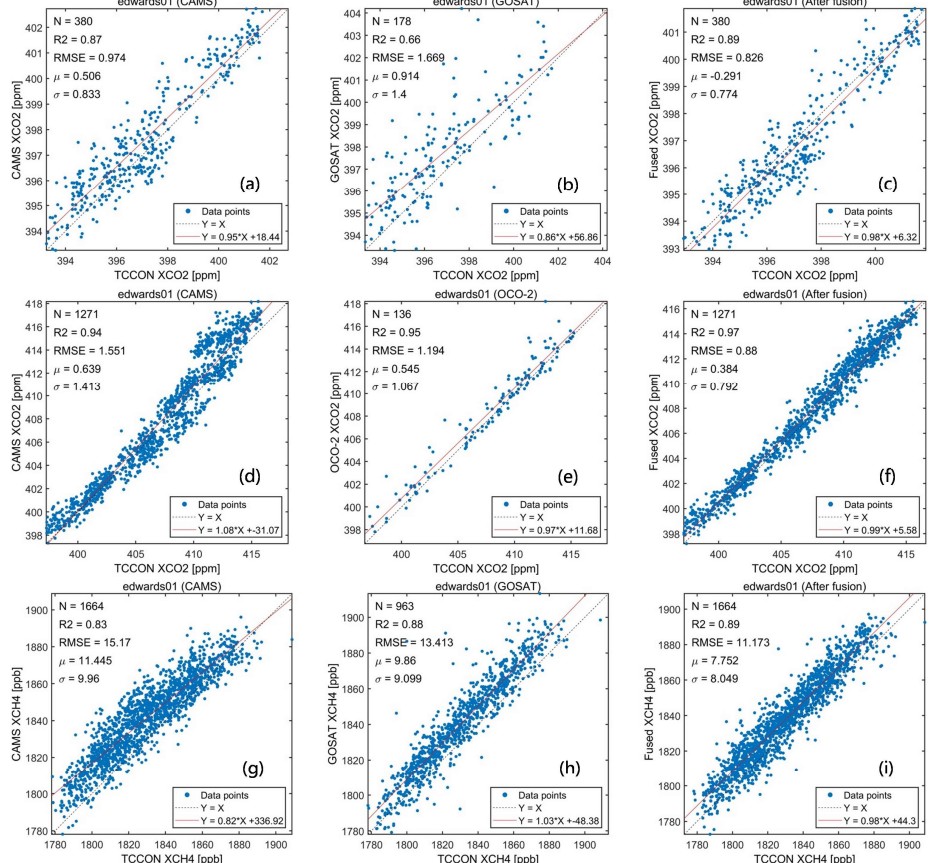

**Figure 4.** Scatter-plots of the in-situ validation results for (a, d, and g) CAMS-EGG4, (b and h) GOSAT, (e) OCO-2, and (c, f, and i) fused results on edwards01. Black dotted and red full lines stand for the 1:1 and fitted lines, respectively. X: TCCON data; Y: CAMS-EGG4/GOSAT/OCO-2/fused data. Unit: ppm/ppb to $XCO_2$/$XCH_4$ for RMSE, $\mu$, and $\sigma$.

## 4 Experiment results and discussions

### 4.1 Overall in-situ validation

As displayed in Fig. 2, the $XCO_2$ and $XCH_4$ measurements from 29 TCCON in-situ stations are adopted for the validation, which evenly distribute over the globe. In this study, TCCON measurements of ± 1 hour on the satellite overpass times (~

13:00 and 13:36 local time, see Table 2) are co-matched with the CAMS-EGG4/GOSAT/OCO-2/fused data around each station
with a diameter of 2°. Figure 3 depicts the overall in-situ validation results for the CAMS-EGG4, GOSAT, OCO-2, and fused
results. The amounts of data points (N) are sufficient (e.g., 1337 for OCO-2 $XCO_2$ and 5402 for GOSAT $XCH_4$) to support the
reliability of validation results.

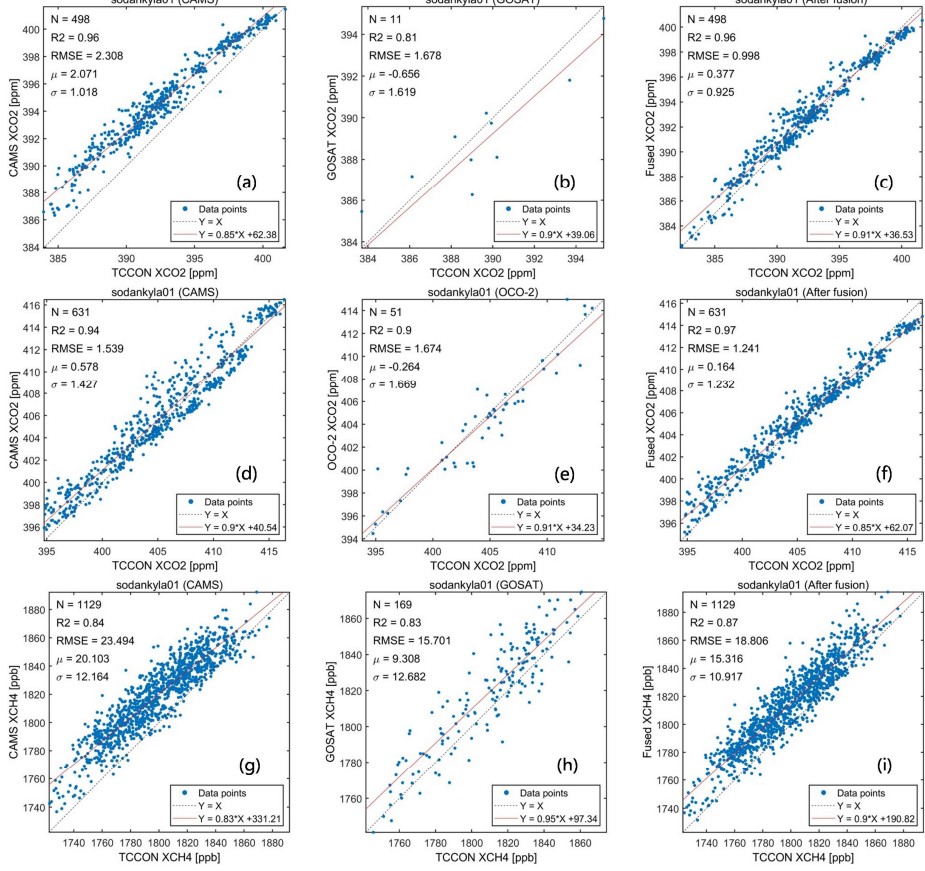

**Figure 5.** Scatter-plots of the in-situ validation results for (a, d, and g) CAMS-EGG4, (b and h) GOSAT, (e) OCO-2, and (c, f, and i) fused
results on sodankyla01. Black dotted and red full lines stand for the 1:1 and fitted lines, respectively. X: TCCON data; Y: CAMS-
EGG4/GOSAT/OCO-2/fused data. Unit: ppm/ppb to $XCO_2$/$XCH_4$ for RMSE, $\mu$, and $\sigma$.
As shown in Fig. 3, the $XCO_2$ from OCO-2 and $XCH_4$ from GOSAT perform better than those from CAMS-EGG4, with larger
$R^2$, smaller RMSE, and smaller $\sigma$. After fusion, the $XCO_2$ (2015-2020) and $XCH_4$ (2010-2020) present a greatly superior
accuracy compared to CAMS-EGG4, of which the RMSE ($\sigma$) improvements are 0.443 (0.444) ppm and 3.752 (1.792) ppb for
$XCO_2$ and $XCH_4$, respectively. Meanwhile, the accuracy of the fused results is higher than and close to those of OCO-2 $XCO_2$
and GOSAT $XCH_4$, respectively. These suggest that the proposed fusion method achieves a satisfactory result. Furthermore,
the performance of $XCO_2$ from GOSAT is similar to that of CAMS-EGG4. However, the fused $XCO_2$ (2010-2014) shows
higher accuracy by comparison with both CAMS-EGG4 and GOSAT, indicating the spatiotemporally local fusion ability of S-
STDCT. In conclusion, our fusion method can successfully fuse the data from CAMS-EGG4 and satellites, which effectively

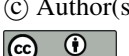



generates GOSAT-like and OCO-2-like values.
**Table 3.** Metrics of the individual in-situ validation results for CAMS-EGG4, GOSAT, and fused XCO₂. The best and second metrics are
denoted with bold and underlined fonts. CE: CAMS-EGG4; AF: after fusion. Unit: ppm for RMSE and $\sigma$.

| Site name | $R^2$ | | | RMSE | | | $\sigma$ | | |
|---|---|---|---|---|---|---|---|---|---|
| | CE | GOSAT | AF | CE | GOSAT | AF | CE | GOSAT | AF |
| bremen01 | 0.91 | 0.85 | **0.92** | 2.810 | 1.732 | **1.533** | 1.376 | 1.757 | **1.189** |
| edwards01 | 0.87 | 0.66 | **0.89** | 0.974 | 1.669 | **0.826** | 0.833 | 1.400 | **0.774** |
| fourcorners01 | 0.88 | **0.91** | 0.86 | 1.237 | 0.867 | **0.844** | 0.848 | **0.590** | 0.801 |
| garmisch01 | 0.91 | 0.86 | **0.93** | 2.141 | 1.575 | **1.070** | 1.275 | 1.592 | **1.067** |
| jpl02 | 0.89 | 0.86 | **0.90** | 1.535 | 1.299 | **1.075** | 0.961 | 1.299 | **0.918** |
| saga01 | 0.90 | 0.91 | **0.93** | 1.362 | 1.494 | **1.333** | 1.313 | 1.201 | **1.065** |
| lauder02 | 0.83 | 0.70 | **0.87** | 0.584 | 1.095 | **0.606** | **0.585** | 1.088 | 0.600 |
| lamont01 | 0.79 | **0.88** | **0.88** | 1.928 | 0.986 | **0.976** | 1.327 | **0.973** | 0.976 |
| orleans01 | 0.89 | 0.75 | **0.91** | 2.105 | 1.666 | **0.964** | 1.144 | 1.440 | **0.964** |
| parkfalls01 | 0.92 | 0.86 | **0.93** | 2.088 | 1.703 | **1.138** | 1.309 | 1.697 | **1.137** |
| pasadena01 | 0.70 | 0.74 | **0.75** | **1.260** | 1.296 | 1.642 | 1.261 | 1.287 | **1.177** |
| sodankyla01 | **0.96** | 0.81 | **0.96** | 2.308 | 1.678 | **0.998** | 1.018 | 1.619 | **0.925** |
| tsukuba02 | 0.80 | **0.82** | 0.78 | **1.179** | 1.651 | 1.494 | **1.157** | 1.263 | 1.202 |

**Table 4.** Metrics of the individual in-situ validation results for CAMS-EGG4, OCO-2, and fused XCO₂. The best and second metrics are
denoted with bold and underlined fonts. CE: CAMS-EGG4; AF: after fusion. Unit: ppm for RMSE and $\sigma$.

| Site name | $R^2$ | | | RMSE | | | $\sigma$ | | |
|---|---|---|---|---|---|---|---|---|---|
| | CE | OCO-2 | AF | CE | OCO-2 | AF | CE | OCO-2 | AF |
| bremen01 | 0.91 | **0.99** | 0.93 | 1.718 | **1.126** | 1.476 | 1.678 | **1.066** | 1.459 |
| burgos01 | 0.91 | **0.95** | 0.94 | 1.324 | **0.715** | 0.933 | 1.144 | **0.709** | 0.823 |
| edwards01 | 0.94 | 0.95 | **0.97** | 1.551 | 1.194 | **0.880** | 1.413 | 1.067 | **0.792** |
| easttroutlake01 | 0.92 | 0.87 | **0.94** | 1.334 | 1.802 | **1.195** | 1.303 | 1.812 | **1.196** |
| eureka01 | 0.94 | 0.93 | **0.97** | 2.081 | 2.224 | **1.427** | 1.436 | 1.555 | **1.171** |
| garmisch01 | 0.91 | 0.93 | **0.96** | 1.586 | 1.569 | **1.019** | 1.579 | 1.354 | **1.010** |
| hefei01 | 0.88 | **0.97** | 0.91 | 1.447 | **1.163** | 1.283 | 1.450 | **0.735** | 1.192 |
| izana01 | 0.96 | 0.88 | **0.99** | 1.215 | 1.413 | **0.576** | 1.209 | 1.417 | **0.555** |
| jpl02 | 0.75 | **0.89** | 0.76 | 2.151 | **1.146** | 1.525 | 1.221 | **0.885** | 1.174 |
| saga01 | 0.89 | **0.95** | 0.94 | 1.890 | **1.087** | 1.263 | 1.873 | **1.090** | 1.254 |
| karlsruhe01 | 0.89 | **0.93** | **0.93** | 1.747 | **1.327** | 1.375 | 1.749 | **1.318** | 1.376 |
| lauder02 | 0.96 | 0.89 | **0.97** | 1.213 | 1.000 | **0.492** | 0.518 | 0.993 | **0.469** |
| lauder03 | **0.94** | 0.72 | **0.94** | 1.288 | 1.064 | **0.565** | 0.863 | 1.070 | **0.538** |
| nicosia01 | 0.79 | 0.91 | **0.94** | 2.319 | **0.731** | 0.862 | 1.133 | 0.661 | **0.641** |
| nyalesund01 | 0.94 | 0.93 | **0.97** | 1.942 | 2.233 | **1.664** | 1.573 | 1.707 | **1.446** |
| lamont01 | 0.92 | **0.97** | 0.96 | 1.505 | **0.956** | 0.964 | 1.489 | **0.794** | 0.929 |
| orleans01 | 0.92 | 0.93 | **0.96** | 1.450 | 1.144 | **1.108** | 1.361 | 1.121 | **1.007** |
| parkfalls01 | 0.93 | **0.96** | 0.95 | 1.518 | 1.210 | **1.160** | 1.518 | 1.211 | **1.160** |
| pasadena01 | 0.91 | 0.93 | **0.95** | 1.689 | 1.543 | **1.382** | 1.581 | 1.329 | **1.160** |
| paris01 | 0.89 | 0.92 | **0.93** | 1.910 | **1.418** | 1.451 | 1.867 | **1.433** | 1.437 |
| reunion01 | 0.96 | **0.97** | **0.97** | 1.276 | 0.878 | **0.874** | 0.827 | 0.886 | **0.812** |
| rikubetsu01 | 0.90 | **0.96** | 0.93 | 1.688 | **1.023** | 1.320 | 1.667 | **1.033** | 1.293 |
| sodankyla01 | 0.94 | 0.90 | **0.97** | 1.539 | 1.674 | **1.241** | 1.427 | 1.669 | **1.232** |
| tsukuba02 | 0.92 | **0.94** | 0.93 | 1.429 | **1.169** | 1.276 | 1.322 | **1.134** | 1.265 |
| xianghe01 | 0.61 | **0.89** | 0.73 | 2.513 | **1.411** | 1.960 | 2.487 | **1.430** | 1.959 |




**Table 5.** Metrics of the individual in-situ validation results for CAMS-EGG4, GOSAT, and fused XCH4. The best and second metrics are denoted with bold and underlined fonts. CE: CAMS-EGG4; AF: after fusion. Unit: ppb for RMSE and $\sigma$.

| Site name | $R^2$ | | | RMSE | | | $\sigma$ | | |
|---|---|---|---|---|---|---|---|---|---|
| | CE | GOSAT | AF | CE | GOSAT | AF | CE | GOSAT | AF |
| bremen01 | 0.84 | **0.90** | 0.87 | 19.397 | 15.328 | **14.969** | 12.507 | **9.868** | 10.938 |
| burgos01 | 0.80 | **0.89** | **0.89** | 10.981 | 10.455 | **8.096** | 9.194 | **6.136** | 7.216 |
| edwards01 | 0.83 | 0.88 | **0.89** | 15.170 | 13.413 | **11.173** | 9.960 | 9.099 | **8.049** |
| fourcorners01 | 0.40 | **0.71** | 0.51 | 14.732 | **7.714** | 9.847 | 9.711 | **6.710** | 8.777 |
| garmisch01 | 0.83 | 0.85 | **0.89** | 16.693 | 13.258 | **12.267** | 11.568 | 11.643 | **9.577** |
| hefei01 | 0.54 | 0.56 | **0.66** | 22.072 | **15.377** | 16.814 | 16.165 | **13.370** | 13.826 |
| jpl02 | 0.81 | **0.88** | 0.86 | 16.989 | **9.679** | 9.788 | 11.288 | **8.840** | 9.604 |
| saga01 | 0.85 | **0.92** | 0.89 | 11.299 | **9.089** | 9.311 | 10.091 | **8.422** | 9.147 |
| karlsruhe01 | 0.70 | **0.80** | 0.81 | 13.688 | 11.913 | **10.042** | 11.564 | 11.370 | **9.177** |
| lauder02 | 0.66 | **0.84** | 0.65 | 18.460 | **8.632** | 11.323 | 11.390 | **6.923** | 10.189 |
| lauder03 | 0.46 | **0.76** | 0.57 | 16.568 | **8.531** | 12.166 | 10.965 | **6.491** | 9.347 |
| lamont01 | 0.82 | **0.94** | 0.88 | 11.762 | 12.204 | **9.497** | 11.494 | **7.015** | 9.460 |
| orleans01 | 0.80 | **0.88** | **0.88** | 18.341 | 13.734 | **13.305** | 12.038 | 9.690 | **9.395** |
| parkfalls01 | 0.79 | **0.87** | 0.84 | 17.107 | 14.892 | **13.784** | 13.396 | **10.548** | 11.519 |
| pasadena01 | 0.82 | **0.90** | 0.88 | 12.658 | **8.396** | 8.845 | 10.544 | **8.094** | 8.802 |
| paris01 | 0.75 | 0.73 | **0.84** | 12.313 | 13.077 | **9.578** | 10.319 | 11.437 | **8.383** |
| reunion01 | 0.51 | 0.41 | **0.73** | 18.245 | 13.846 | **10.092** | 10.221 | 11.427 | **7.432** |
| rikubetsu01 | 0.60 | **0.81** | 0.72 | 21.166 | 20.160 | **18.250** | 15.263 | **11.481** | 12.759 |
| sodankyla01 | 0.84 | 0.83 | **0.87** | 23.494 | **15.701** | 18.806 | 12.164 | 12.682 | **10.917** |
| tsukuba02 | 0.77 | **0.86** | 0.83 | 11.726 | **8.165** | 8.704 | 9.401 | **7.623** | 8.424 |
| xianghe01 | 0.63 | **0.69** | 0.63 | **14.851** | 15.840 | 15.266 | 14.734 | **13.752** | 14.736 |

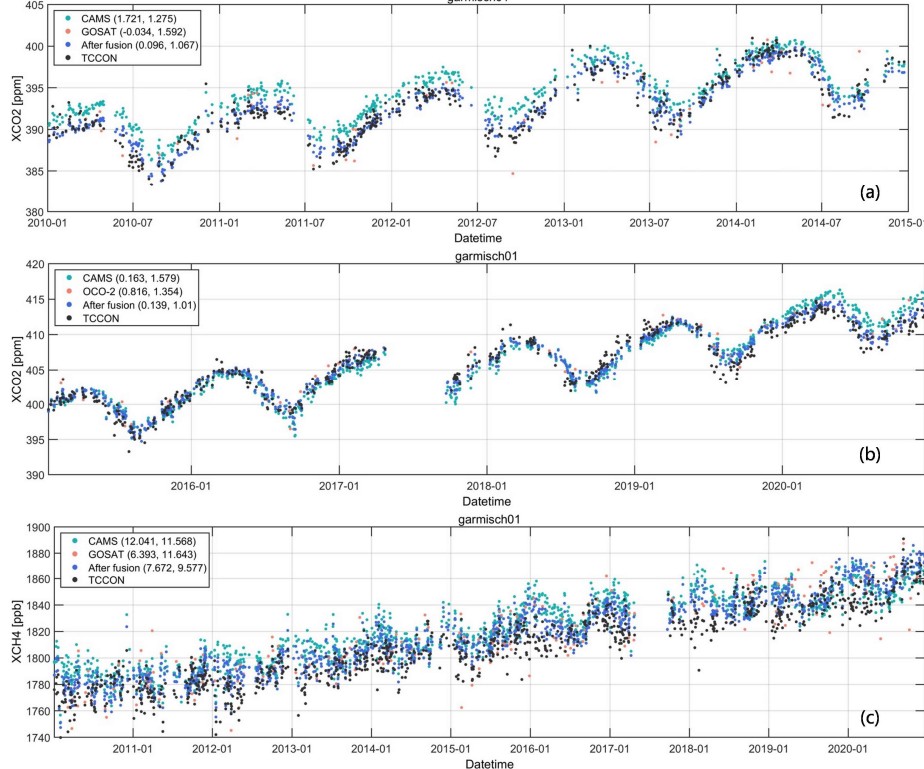

**Figure 6.** Scatter-plots of the time series for daily CAMS-EGG4, GOSAT, OCO-2, fused, and TCCON data on garmisch01. The first and second numbers in the bracket represent $\mu$ and $\sigma$, respectively. Unit: ppm/ppb to XCO2/XCH4 for $\mu$ and $\sigma$.



## 4.2 Individual in-situ validation and time series

Figure 4, 5, and S1-S24 illustrate the individual in-situ validation results for the CAMS-EGG4, GOSAT, OCO-2, and fused results on each TCCON in-situ station. It is worth noting that only the stations where the individual validation results are significant (p-level < 0.01) for all datasets (i.e., CAMS-EGG4, GOSAT, OCO-2, and the fused results) are presented. Since the space of text is limited, two stations named edwards01 and sodankyla01 are selected as examples (see Fig. 4 and 5), which locate in North America and Europe, respectively. As can be seen, the fused results achieve the best performance compared to CAMS-EGG4, GOSAT, and OCO-2 on edwards01 and sodankyla01, with the $R^2$ ranging from 0.87 to 0.97. Especially, the large overestimation of $XCO_2$ for CAMS-EGG4 on sodankyla01 ($\mu$ = 2.071 ppm) is well mitigated after fusion ($\mu$ = 0.377 ppm), even for the poor data availability of GOSAT (N = 11). This indicates the strong universality of the proposed fusion method. The valid individual validation results on all stations are given in Table 3-5 (more details in the Supplement, Fig. S1-S24). It can be observed that the performance of the fused results exceeds those of CAMS-EGG4 and GOSAT/OCO-2 for almost all stations and ~ 70 % of stations, respectively.

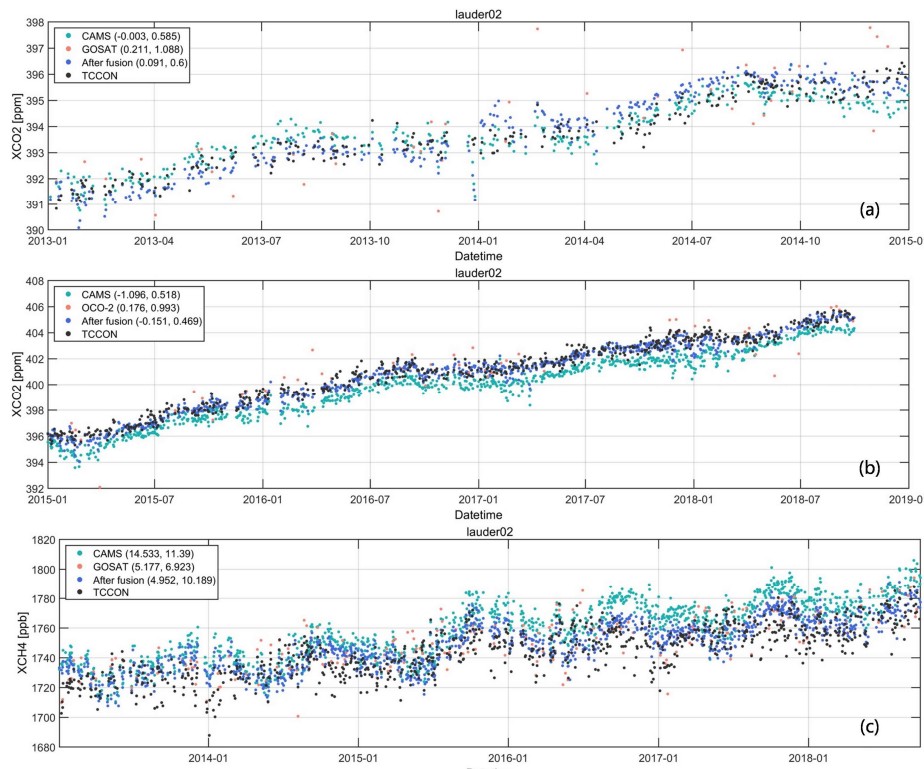

**Figure 7.** Scatter-plots of the time series for daily CAMS-EGG4, GOSAT, OCO-2, fused, and TCCON data on lauder02. The first and second numbers in the bracket represent $\mu$ and $\sigma$, respectively. Unit: ppm/ppb to $XCO_2$/$XCH_4$ for $\mu$ and $\sigma$.





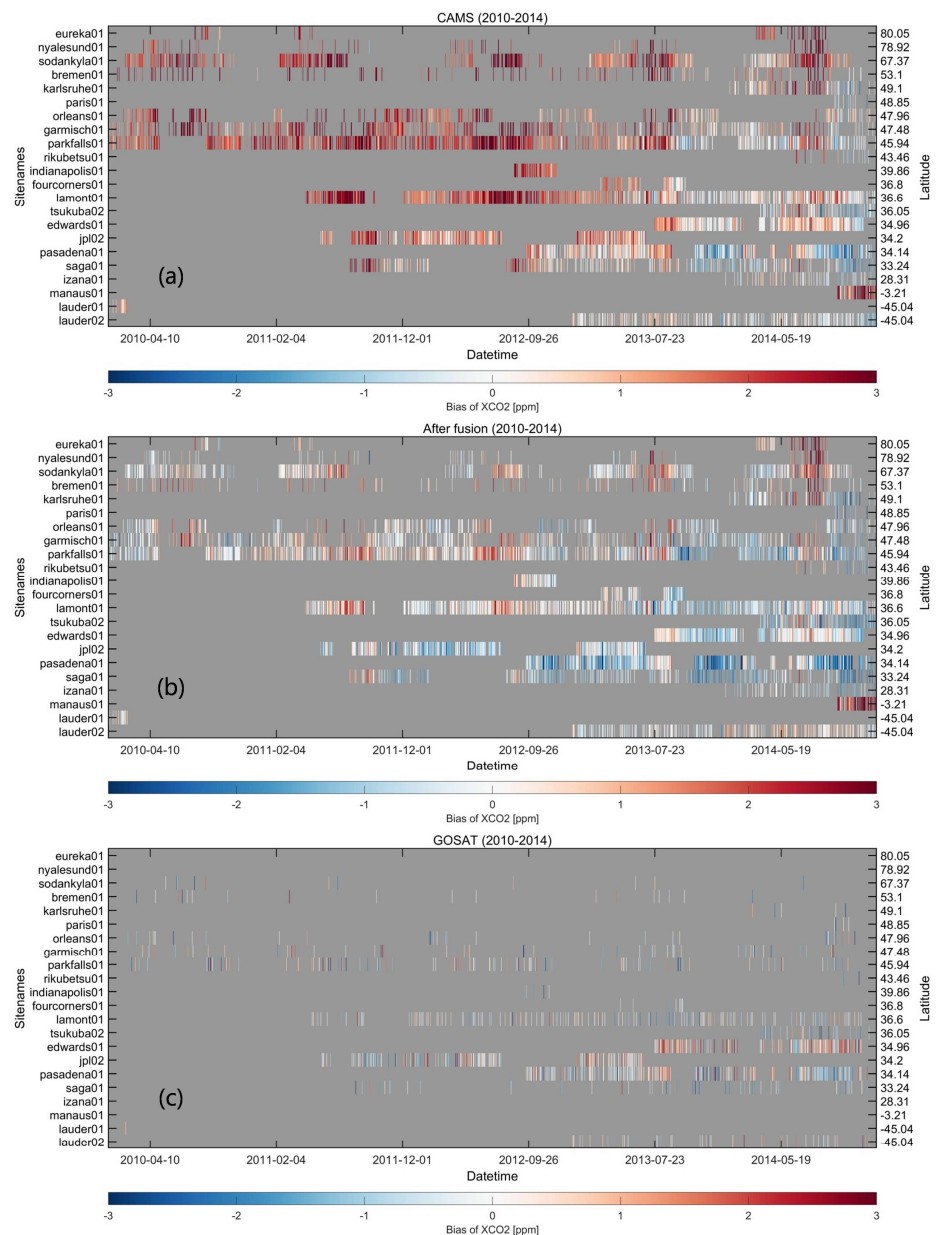

**Figure 8.** Heat maps of the biases between daily (a) CAMS-EGG4/(b) fused/(c) GOSAT and TCCON XCO$_2$ over time and latitude. Color ramps stand for the biases of XCO$_2$. Background colors (grey) indicate the missing data.



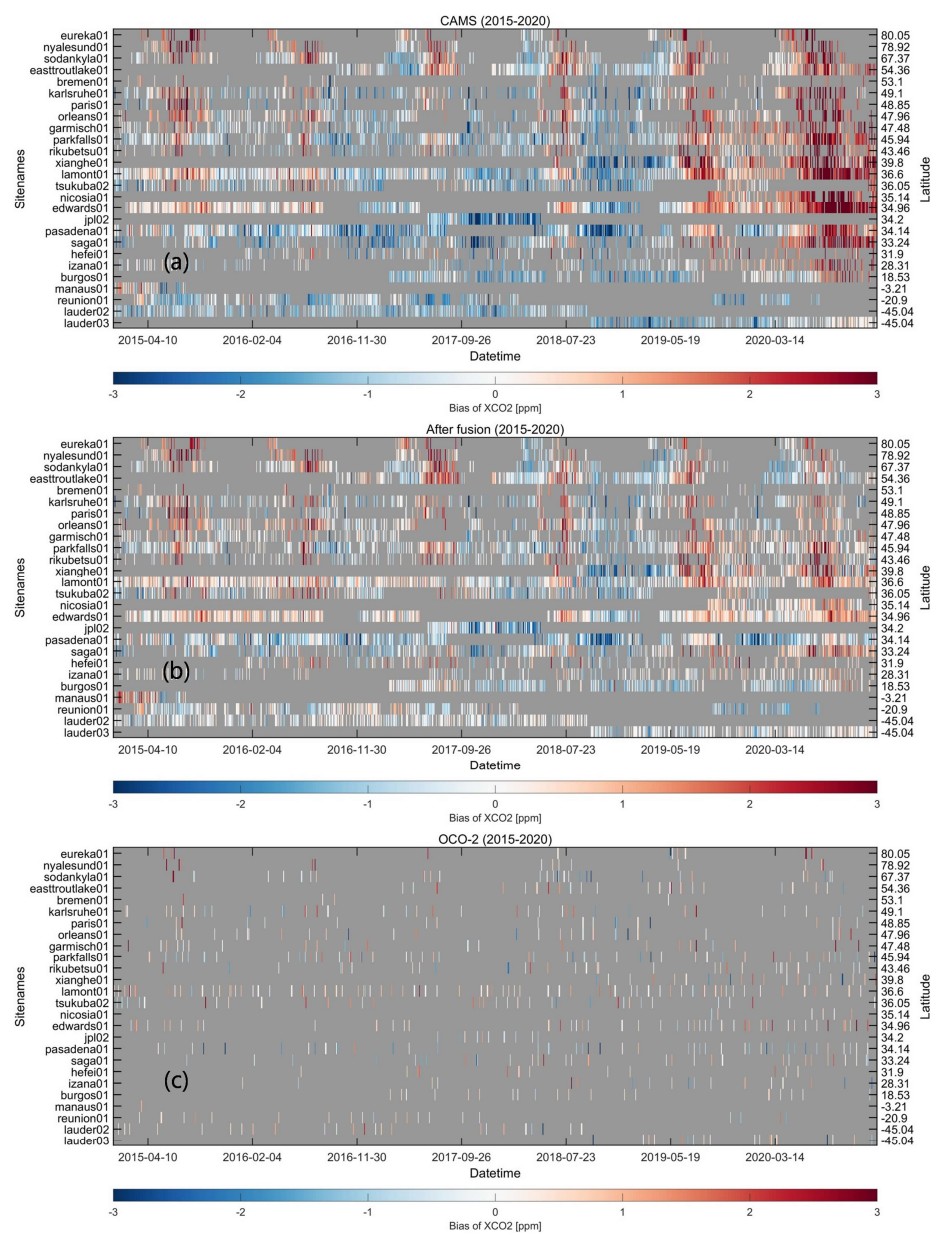

**Figure 9.** Heat maps of the biases between daily (a) CAMS-EGG4/(b) fused/(c) OCO-2 and TCCON XCO$_2$ over time and latitude. Color ramps stand for the biases of XCO2. Background colors (grey) indicate the missing data.

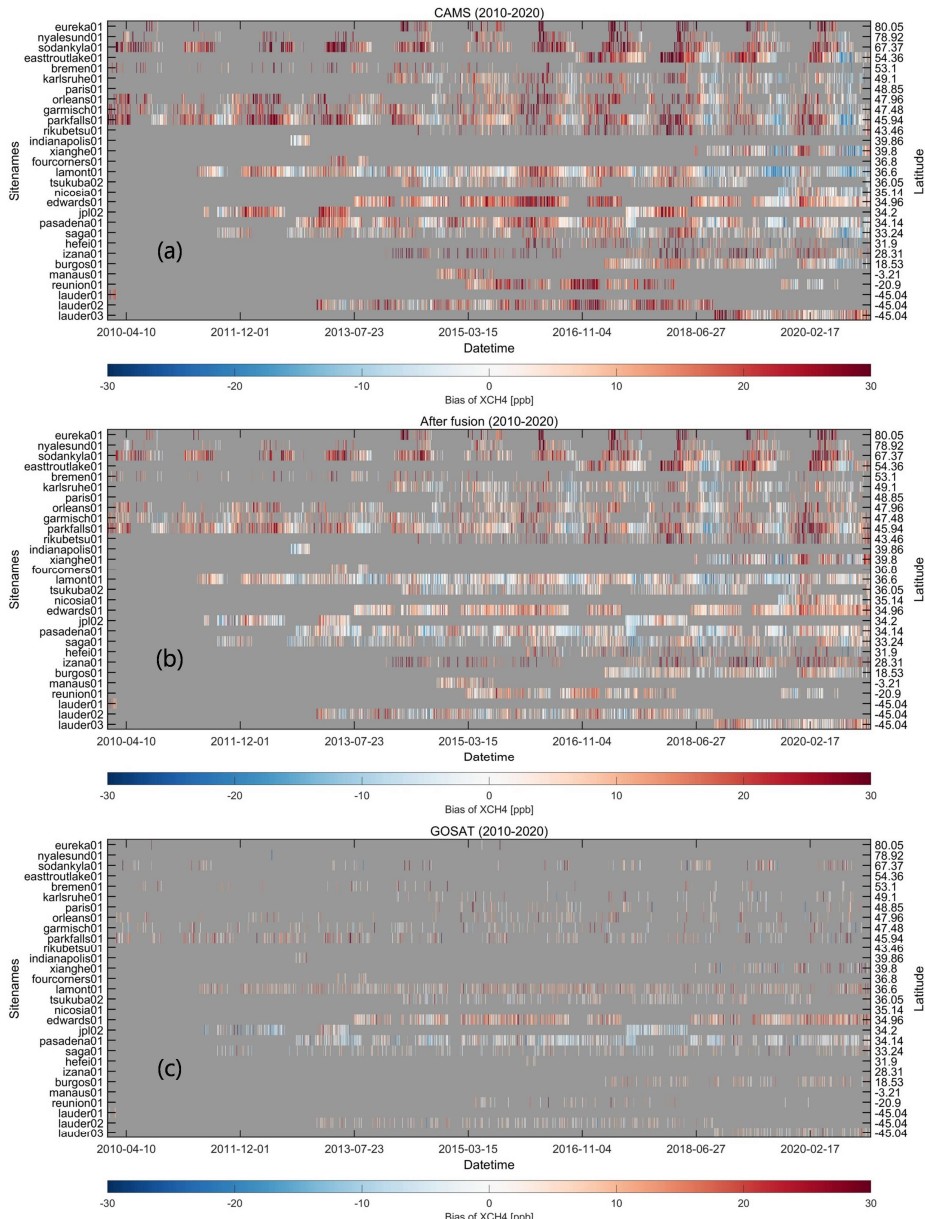

**Figure 10.** Heat maps of the biases between daily (a) CAMS-EGG4/(b) fused/(c) GOSAT and TCCON XCH₄ over time and latitude. Color ramps stand for the biases of XCO2. Background colors (grey) indicate the missing data.

Figure 6, 7, and S25-S48 demonstrate the time series for daily CAMS-EGG4, GOSAT, OCO-2, fused, and TCCON data on each in-situ station. Similarly, two stations, i.e., garmisch01 and lauder02, are regarded as examples, which locate in Europe and Oceania, respectively. As depicted in Fig. 6, the $XCO_2$ from CAMS-EGG4 is markedly overestimated on garmisch01 from 2010 to 2014 and in 2020. After fusion, the $XCO_2$ presents an equal trend compared to TCCON measurements over time, with smaller $\mu$ (0.096 and 0.139 ppm) and $\sigma$ (1.067 and 1.01 ppm). In the meantime, the overestimation of CAMS-EGG4 XCH₄



is also mitigated on garmisch01 through our fusion method. Regarding lauder02, Figure 7 shows that CAMS-EGG4 generates
underestimated XCO₂ (2015-2019) and overestimated XCH₄. The $\mu$ and $\sigma$ of the fused results (e.g., 4.952 and 10.189 ppb
for XCH₄) are significantly improved on lauder02. The time series on other stations are provided in the Supplement (see Fig.
S25-S48). The readers can refer to them if interested, which will not be further described here.

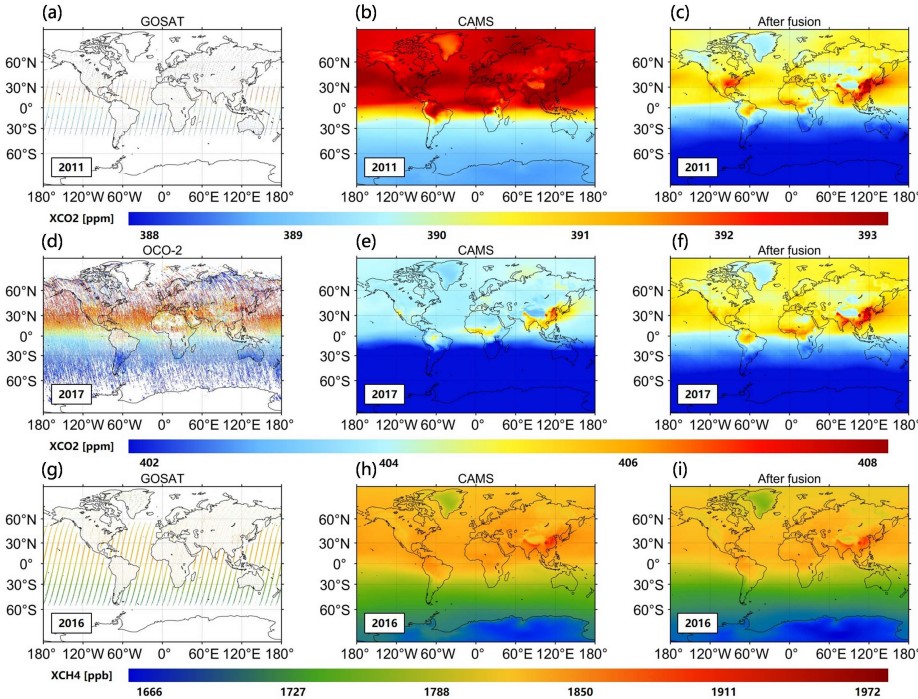

**Figure 11.** Annual (a and g) GOSAT, (d) OCO-2, (b, e, and h) CAMS-EGG4, and (c, f, and i) fused XCO₂/XCH₄ over the globe. Color
ramps stand for the values of XCO₂ and XCH₄.

**4.3 Uncertainty analyses**

Figure 8-10 display the biases between daily CAMS-EGG4/fused/GOSAT/OCO-2 and TCCON data over time and latitude.
As observed in Fig. 8 and 9, a large overestimation generally exists in the CAMS-EGG4 XCO₂ from 2010 to 2014 and in 2020,
especially before 2013 and in 2020 (> 3 ppm). These are attributed to the considerable errors in the satellite data assimilated
(2010-2014) and that anthropogenic emissions are not modified for COVID-19 lockdowns in 2020 (Agusti-Panareda et al.,
2022). After fusion, the biases of XCO₂ are well improved for most TCCON in-situ stations from 2010 to 2014 and in 2020,
whose patterns are similar to those of GOSAT and OCO-2 XCO₂, respectively. This indicates that the proposed fusion method
can effectively correct the biases in CAMS-EGG4 due to the issues from assimilation data. Meanwhile, CAMS-EGG4
generates distinctly underestimated XCO₂ from 2016 to 2019 on the stations of latitude < 40° N, which is also mitigated via
the S-STDCT fusion method (see Fig. 10). Moreover, the CAMS-EGG4 XCH₄ frequently presents a large positive bias (> 30
ppb), while the fused XCH₄ only enhances the performance on the stations of latitude < 50° N. The improvements for other



stations require our further efforts in the future.

### 4.4 Assessment of spatial distribution on multi-temporal scales

Figure 11 demonstrates the comparisons of annual GOSAT, OCO-2, CAMS-EGG4, and fused $XCO_2/XCH_4$ over the globe. A
total of three years are selected, including 2011, 2017, and 2016. As can be seen, the fused results present coincident spatial
patterns with GOSAT and OCO-2, even if the annual GOSAT and OCO-2 data are greatly sparse. Particularly, the large
overestimation and underestimation of CAMS-EGG4 $XCO_2$ in 2011 and 2017 are significantly modified after fusion,
respectively, which are mutually confirmed with the descriptions in Section 4.3.
Figure 12 depicts the multi-year mean fused global $XCO_2$ and $XCH_4$ from 2010 to 2020. Generally, the spatial patterns of
$XCO_2$ and $XCH_4$ are divided by the equator. The high values of $XCO_2$ and $XCH_4$ mainly distribute over Asia, e.g., China and
India, which is attributed to the large anthropogenic emissions (Kenea et al., 2023; Liu et al., 2020; Turner et al., 2015;
Hotchkiss et al., 2015). In the meantime, considerable natural emissions, e.g., wildfires (Arora and Melton, 2018), also can
obviously increase the $XCO_2$ values, such as in central Africa and northern South America. Figure 13 and 14 illustrate the
seasonal fused $XCO_2$ and $XCH_4$ from 2010 to 2020 over the globe, respectively. As displayed, seasonal changes of global
$XCO_2$ and $XCH_4$ spatial patterns are clearly reflected in the fused results. Compared to $XCH_4$, the global spatial patterns of
$XCO_2$ vary more drastically. This is likely driven by the spatiotemporal heterogeneity of meteorological fields (Liu et al., 2011)
and different emission sources of $CO_2$ and $CH_4$.

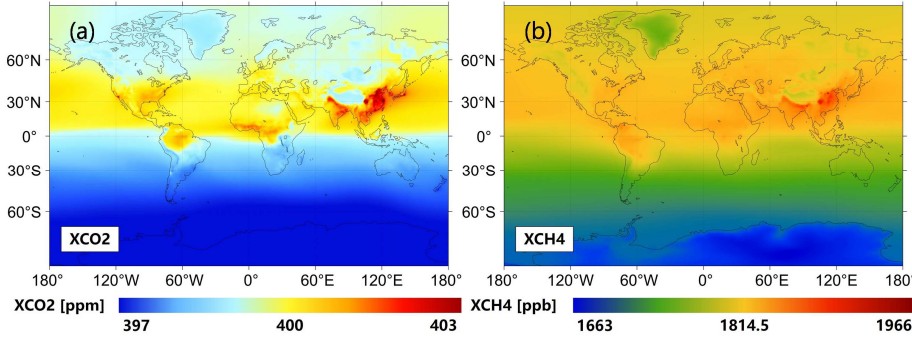


**Figure 12.** Multi-year mean fused (a) $XCO_2$ and (b) $XCH_4$ from 2010 to 2020 over the globe. Color ramps stand for the values of $XCO_2$ and
$XCH_4$.



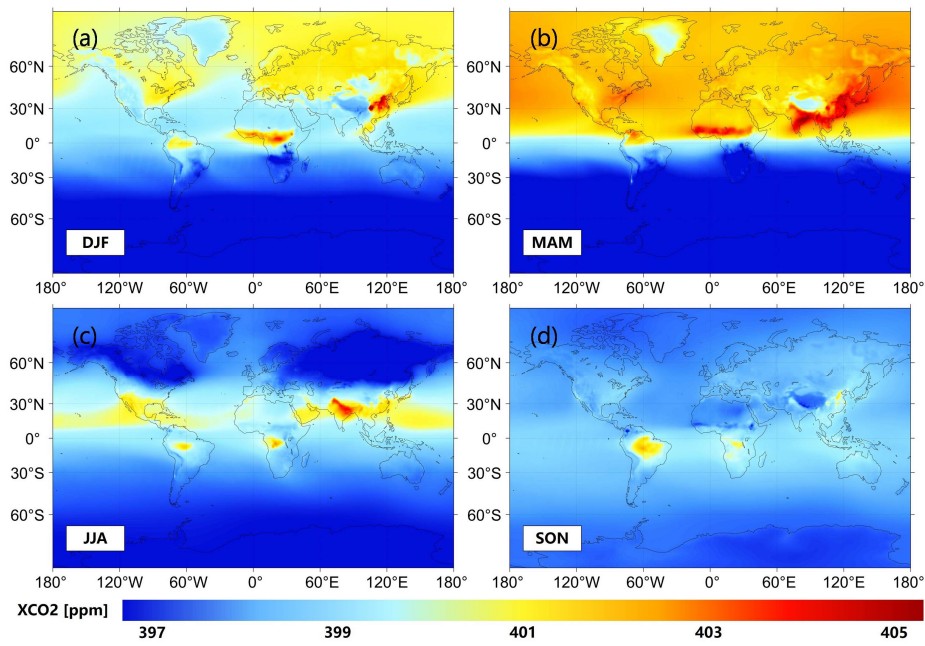


**Figure 13.** Seasonal fused $XCO_2$ from 2010 to 2020 over the globe. The color ramp stands for the value of $XCO_2$. (a) DJF, (b) MAM, (c) JJA, and (d) SON denote Dec. to Feb., Mar. to May., Jun. to Aug., and Sep. to Nov., respectively.

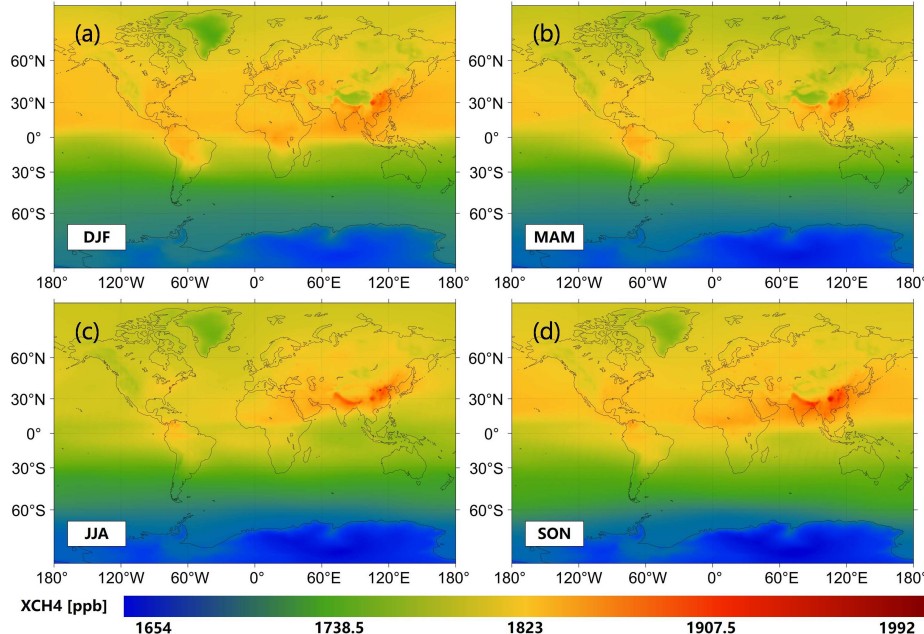


**Figure 14.** Seasonal fused $XCH_4$ from 2010 to 2020 over the globe. The color ramp stands for the value of $XCH_4$. (a) DJF, (b) MAM, (c) JJA, and (d) SON denote Dec. to Feb., Mar. to May., Jun. to Aug., and Sep. to Nov., respectively.

Figure 15 and 16 map the annual fused global $XCO_2$ and $XCH_4$ from 2010 to 2020, respectively, including their trends. As



observed in Fig. 15, the $CO_2$ levels continuously increase from 2010 to 2020, with the mean $XCO_2$ values ranging from $\leq$

386 to $\geq$ 416 ppm. However, the trends of $XCO_2$ only present small spatial differences (~ 0.2 ppm per year), of which the

large growth rates primally distribute along the equator, especially for China ($\geq$ 2.5 ppm per year). It is worth noting that the

growth rates of $XCO_2$ are relatively slight ($\leq$ 2.3 ppm per year) in northern South America compared to other regions. This is

likely caused by the effects from the carbon sequestration of forests (Chazdon et al., 2016). Besides, the $XCH_4$ values also

notably rise from 2010 to 2020, of which the maximum is not less than 2008 ppb in 2020 (see Fig. 16). The large growth rates

of $XCH_4$ are majorly discovered over southern Asia and northern Europe.

## 5 Data availability

The fused results can be freely accessed at http://doi.org/10.5281/zenodo.7388893 (Wang et al., 2022b). The daily global

seamless gridded (0.25°) $XCO_2$ and $XCH_4$ from 2010 to 2020 are stored in the netCDF4 format with a file size of ~ 3.5 MB

for each day. The units of $XCO_2$ and $XCH_4$ are ppm and ppb, respectively.

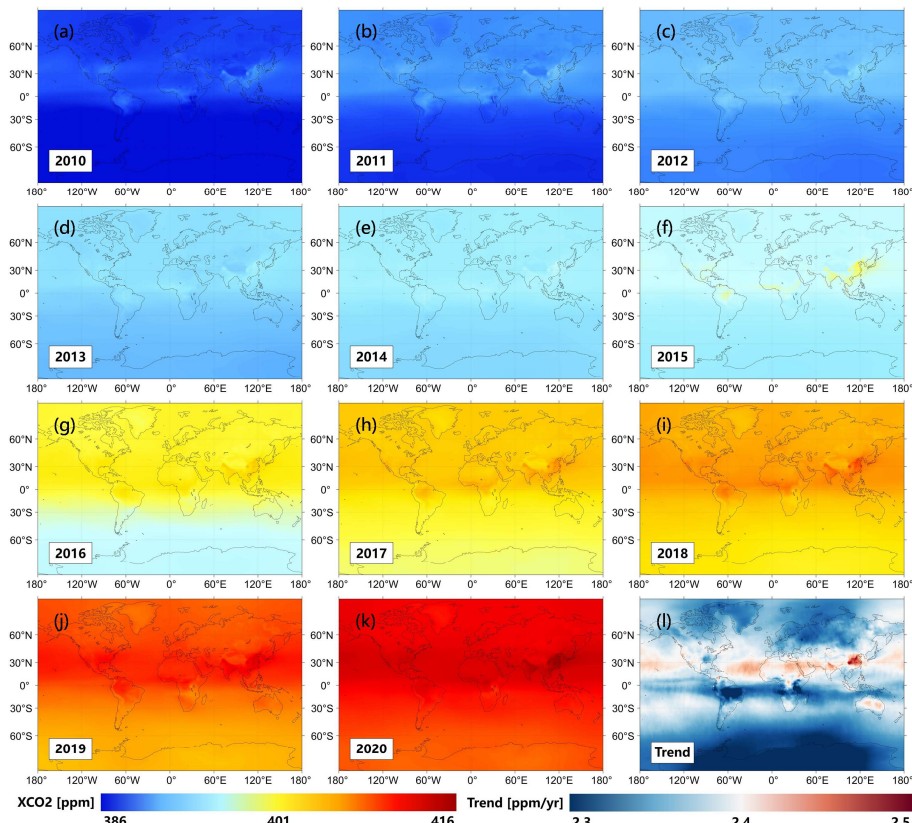

**Figure 15.** Annual fused (a-k) $XCO_2$ and (l) its trend from 2010 to 2020 over the globe. Color ramps stand for the values of $XCO_2$ and its
trend. ppm/yr: ppm per year.



## 6 Conclusions

In our study, a novel spatiotemporally self-supervised fusion method, i.e., S-STDCT, is proposed to acquire long-term daily seamless globally distributed $XCO_2$ and $XCH_4$ products from 2010 to 2020 at the grids of 0.25°. A total of three datasets are adopted, which include GOSAT, OCO-2, and CAMS-EGG4. Since the data from GOSAT and OCO-2 is greatly sparse in space-time domain, the algorithm for frequency domain (the *STDCT*) is applied in the fusion task. Validation results show that the S-STDCT fusion method performs well over the globe, with the $\sigma$ ($R^2$) of ~ 1.18 ppm (0.91 or 0.95) and 11.3 ppb (0.9) for $XCO_2$ and $XCH_4$ against TCCON measurements, respectively. Generally, the accuracy of fused results is distinctly superior to that of CAMS-EGG4, which also exceeds or equals those of GSOAT and OCO-2. Particularly, the proposed fusion method effectively modifies the large biases in CAMS-EGG4 caused by the issues from assimilation data, such as the uncorrected anthropogenic emission inventories for COVID-19 lockdowns in 2020. Besides, the spatial patterns of fused results remain coincident with GOSAT and OCO-2, which can accurately display the long-term and seasonal changes of global $XCO_2$ and $XCH_4$ spatial distribution. The long-term (2010-2020) daily global seamless gridded (0.25°) fused results are available at http://doi.org/10.5281/zenodo.7388893 (Wang et al., 2022b).

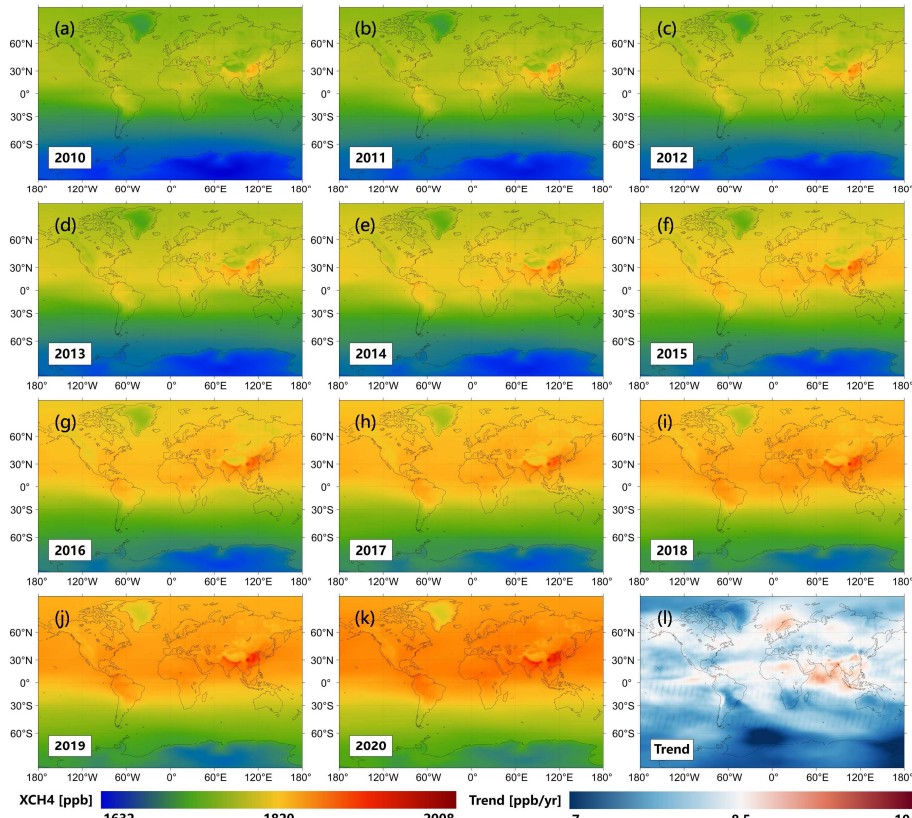

**Figure 16.** Annual fused (a-k) $XCH_4$ and (l) its trend from 2010 to 2020 over the globe. Color ramps stand for the values of $XCH_4$ and its trend. ppb/yr: ppb per year.



**Author contributions**
YW designed the study, collected and processed the data, analyzed the results, and wrote the paper. QQY and TWL provided
constructive comments on the paper. YJY, SQZ, and LPZ revised the paper. All authors contributed to the study.
**Competing interests**
The contact author has declared that none of the authors has any competing interests.
**Disclaimer**
Publisher's note: Copernicus Publications remains neutral with regard to jurisdictional claims in published maps and
institutional affiliations.
**Acknowledgments**
The authors would like to express gratitude to the Goddard Earth Science Data and Information Services Center for providing
the GOSAT and OCO-2 $XCO_2$ products (last access: 20 November 2022 and 27 November 2022), the Centre for
Environmental Data Analysis for providing the GOSAT $XCH_4$ product (last access: 18 November 2022), the Copernicus
Climate Data Store for providing the CAMS-EGG4 $XCO_2$ and $XCH_4$ products (last access: 25 November 2022), the Total
Carbon Column Observing Network (hosted by CaltechDATA at https://tccondata.org; Chair: Dr. Debra Wunch) for
establishing and maintaining in-situ stations (last access: 18 November 2022).
**Financial support**
Our work is supported by the National Natural Science Foundation of China (No. 41922008), the Basic and Applied Basic
Research Foundation of Guangdong Province (No. 2021A1515110567), and the Hubei Science Foundation for Distinguished
Young Scholars (No. 2020CFA051).

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
