# Peer review of "Seamless mapping of long-term (2010-2020) daily global XCO2 and"

_Earth System Science Data, 2023_

## Author Comment (AC1)

**Response to Comments on the Manuscript (essd-2023-28):**

**Seamless mapping of long-term (2010-2020) daily global XCO$_2$ and XCH$_4$ from GOSAT, OCO-2, and CAMS-EGG4 with a spatiotemporally self-supervised fusion method**
* * *
Dear Editors and Referees,

We would like to sincerely express our gratitude to you for your careful reading and constructive comments.

According to the comments, we have tried our best to improve the manuscript, and an item-by-item response follows. The modified parts have been highlighted in yellow color in the revised manuscript.

Once again, we are particularly grateful for your careful reading and constructive comments. Thanks very much for your time.

Best regards,

Qiangqiang Yuan

**Response to Comments of Referee #1:**

**General comments:**

The manuscript designed a self-supervised fusion method based on spatiotemporal Discrete Cosine Transform to fuse sparse satellite data and global full-coverage GHGs reanalysis data for the seamless estimation of long-term daily global $XCO_2$ and $XCH_4$. It is an interesting study, especially for providing long-term $XCO_2$ and $XCH_4$ datasets that don't need any auxiliary variables and analyzing the global pattern of $XCO_2$ and $XCH_4$ trends. I think the paper needs minor revision before acceptance for publication.

**Response:** We would like to take this opportunity to gratefully thank the referee for his/her comments and recommendations for improving the paper. An item-by-item response to the constructive comments follows. Thanks for your time.

**Specific comments:**

**Q1.1:** The assessments of spatial distribution on a daily temporal scale may be necessary for a daily dataset.

**Response:** Thank the referee for his/her valuable comment. The assessments of spatial distribution on a daily temporal scale have been appended in the manuscript.

**The main revision is as follows:**

Figure 12 illustrates the examples of daily fused $XCO_2$ and $XCH_4$ over the globe, consisting of three days in three years. As shown, the fused results display detailed information on atmospheric $CO_2$ and $CH_4$, which clearly indicate their regional and global spatial patterns. In addition, incoherent or factitious spatial distribution is not observed in the fused $XCO_2$ and $XCH_4$. Next, Fig. 13 provides the corresponding daily $XCO_2$ and $XCH_4$ from GOSAT and OCO-2 over the globe. It is worth noting that the daily satellite $XCO_2$ and $XCH_4$ are mapped via footprints due to their significant sparse coverage, which are nearly invisible at grids of 0.25°. As expected, the fused results present identical spatial distribution compared to $XCO_2$ and $XCH_4$ from GOSAT and OCO-2. This suggests the robustness and reliability of the proposed fusion method.

[Figure]

**Figure 12.** Daily fused (a-f) $XCO_2$ and (g-i) $XCH_4$ over the globe. Color ramps stand for the values of $XCO_2$ and $XCH_4$.

[Figure]

**Figure 13.** Daily (a-c) GOSAT, (d-f) OCO-2 $XCO_2$, and (g-i) GOSAT $XCH_4$ over the globe. Color ramps stand for the values of $XCO_2$ and $XCH_4$.

**Q1.2:** **Satellite XCO₂ of 2010-2014 and 2015-2020 come from two datasets. It is crucial for users who rely on the long-term dataset to assess the sensitivity difference between various data. Did the authors compare the difference between the two datasets, and was this difference either ignored or corrected for?**

**Response:** Thank the referee for his/her important comment. Concerning the CAMS-EGG4 product, a total of 7 satellite $XCO_2$ datasets are considered in the assimilation during different overlapped periods (Agusti-Panareda et al., 2022), such as CO2_IAS_NLIS (IASI), CO2_GOS_SRFP (GOSAT), and CO2_GOS_BESD (GOSAT). In our study, the $XCO_2$ from GOSAT and OCO-2 are adopted for the fusion from 2010 to 2020. Table r1 lists the specific information of $XCO_2$ from GOSAT and OCO-2. To compare their fusion performance, Figure r1 depicts the in-situ validation results of the fused results with GOSAT and OCO-2 $XCO_2$ during 2015-2019. As can be seen, the fusion with GOSAT $XCO_2$ will lead to a negative bias of -0.415 ppm and higher RMSE/$\sigma$ compared to OCO-2. This suggests the usage of OCO-2 $XCO_2$ for the fusion after 2015 is more appropriate, which is likely attributed to its better accuracy and larger coverage. Hence, the proposed fusion method utilizes the $XCO_2$ from GOSAT and OCO-2 during 2010-2014 and 2015-2020, respectively. Eventually, the fused results present similar performance during two periods (see Figure r2), indicating the consistency of our long-term $XCO_2$ dataset.

**Table r1.** Specific information of $XCO_2$ from GOSAT and OCO-2.

| Source | Scientific data record | Spatial resolution | Temporal resolution | Available period |
|---|---|---|---|---|
| GOSAT | "XCO2" | 10.5 km (diameter) | Daily (~ 13:00 local time) | 2010-2019 |
| OCO-2 | "XCO2" | 1.29×2.25 km² | Daily (~ 13:36 local time) | 2015-2020 |

[Figure]

**Figure r1.** Density scatter-plots of the in-situ validation results for fused $XCO_2$ with (left) GOSAT and (right) OCO-2 from 2015 to 2019. Black dotted and red full lines stand for the 1:1 and fitted lines, respectively. Color ramps show the normalized densities of data points. X: TCCON data; Y: fused data. Unit: ppm for RMSE, $\mu$, and $\sigma$.

[Figure]

**Figure r2.** Density scatter-plots of the in-situ validation results for (upper) CAMS-EGG4 and (lower) fused $XCO_2$. Black dotted and red full lines stand for the 1:1 and fitted lines, respectively. Color ramps show the normalized densities of data points. X: TCCON data; Y: CAMS-EGG4/fused data. Unit: ppm for RMSE, $\mu$, and $\sigma$.

**Reference:**

Agusti-Panareda, A., Barré, J., Massart, S., Inness, A., Aben, I., Ades, M., Baier, B. C., Balsamo, G., Borsdorff, T., Bousserez, N., Boussetta, S., Buchwitz, M., Cantarello, L., Crevoisier, C., Engelen, R., Eskes, H., Flemming, J., Garrigues, S., Hasekamp, O., Huijnen, V., Jones, L., Kipling, Z., Langerock, B., McNorton, J., Meilhac, N., Noel, S., Parrington, M., Peuch, V.-H., Ramonet, M., Ratzinger, M., Reuter, M., Ribas, R., Suttie, M., Sweeney, C., Tarniewicz, J., and Wu, L.: Technical note: The CAMS greenhouse gas reanalysis from 2003 to 2020, EGUsphere, 1–51, https://doi.org/10.5194/egusphere-2022-283, 2022.

**Q1.3: I have some questions that are not addressed in the Methodology section. What are the range and meaning of M, N, and P in line 175? while u, v and w have the same range as i, j and**

**t. what's the difference and relationship between them?**

**Response:** Thank the referee for his/her careful comment. In our study, $M$, $N$, and $P$ represent the counts of rows (latitude), columns (longitude), and temporal sequences (days), which equal 721 (0.25°, global grids), 1441 (0.25°, global grids), and days of a year (365 or 366), respectively. Besides, $i$, $j$, and $t$ stand for the row, column, and temporal sequence, respectively. By contrast, $u$, $v$, and $w$ denote the transformed coordinates in frequency domain. Although $u$, $v$, and $w$ share the same ranges with $i$, $j$, and $t$, their meanings are different. An example is provided in Figure r3 to show the difference and relationship between them. More related descriptions have been appended in the manuscript.

[Figure]

**Figure r3.** Comparison between CAMS-EGG4 XCO$_2$ and its transformed three-dimensional tensor after *STDCT* in 2015.

**The main revision is as follows:**

$M$, $N$, and $P$ stand for the counts of rows (latitude), columns (longitude), and temporal sequences (days), which equal 721 (0.25°, global grids), 1441 (0.25°, global grids), and days of a year (365 or 366), respectively.

**Q1.4:** **In the Equation 4, $\delta$ represents a series of values or a three-dimensional tensor. Based on Equations 5 and 6, it appears that $\delta$ is a tensor. How are positions without values in $\delta$ handled and does it affect the final iteration process?**

**Response:** Thank the referee for his/her comment. $\delta$ is defined as a three-dimensional tensor, of which the values are only available when satellite XCO$_2$/XCH$_4$ products are valid. In our study, the proposed fusion method can reconstruct the missing information in $\delta$ using its available values. Specific procedures are as follows:

(1) Initialize missing values in $\delta$ through the spatiotemporal nearest neighbor interpolation.

(2) Update $\delta$ by iterations with Eq. (6) based on the spatiotemporal knowledge of self-correlation. Retain the original available values of $\delta$ after each iteration.

(3) Repeat procedure (2) until the number of total iterations or error reaches a predetermined threshold. A schematic diagram to demonstrate the iteration process of $\delta$ (fusion with OCO-2 XCO₂) in 2017-04-10 is depicted in Figure r4.

[Figure]

**Figure r4.** Schematic diagram to show the iteration process of $\delta$ (fusion with OCO-2 XCO₂) in 2017-04-10.

**Q1.5: I am curious that the configuration of the parameters in line 208. How do these parameters affect the results, especially for $\varepsilon$?**

**Response:** Thank the referee for his/her comment. The number of total iterations affects the performance of fused results. The fusion performance firstly increases as the number of total iterations rises and then tends to be relatively stable. $\gamma$ is a relaxation factor to accelerate convergence. The convergence will be faster as $\gamma$ grows up. However, a large $\gamma$ may magnify the error of convergence. $\varepsilon$ represents a smoothing factor. A large $\gamma$ value could result in the loss of high-frequency components (Garcia, 2010). This parameter generally needs to be smaller and smaller to reduce the influence of smoothing during the iterations.

**Q1.10:** Figures: Modify the superscript of $R^2$ and subscript of $XCO_2$ and $XCH_4$ in all Figures.

**Response:** Thank the referee for his/her careful comment. The superscript of $R^2$ and subscript of $XCO_2$ and $XCH_4$ have been modified in all figures.

**The main revision is as follows:**

[Figure]

**Figure 1.** An example of daily spatial footprints for (a) GOSAT $XCO_2$, (b) OCO-2 $XCO_2$, and (c) GOSAT $XCH_4$. Red points signify the available data. Background maps are naturally shaded reliefs over the globe.

[Figure]

**Figure 3.** Density scatter-plots of the in-situ validation results for (a, d, and g) CAMS-EGG4, (b and h) GOSAT, (e) OCO-2, and (c, f, and i) fused results. Black dotted and red full lines stand for the 1:1 and fitted lines, respectively. Color ramps show the normalized densities of data points. X: TCCON data; Y: CAMS-EGG4/GOSAT/OCO-2/fused data. Unit: ppm/ppb to $XCO_2$/$XCH_4$ for RMSE, $\mu$, and $\sigma$.

[Figure]

**Figure 4.** Scatter-plots of the in-situ validation results for (a, d, and g) CAMS-EGG4, (b and h) GOSAT, (e) OCO-2, and (c, f, and i) fused results on edwards01. Black dotted and red full lines stand for the 1:1 and fitted lines, respectively. X: TCCON data; Y: CAMS-EGG4/GOSAT/OCO-2/fused data. Unit: ppm/ppb to $XCO_2$/$XCH_4$ for RMSE, $\mu$, and $\sigma$.

[Figure]

**Figure 5.** Scatter-plots of the in-situ validation results for (a, d, and g) CAMS-EGG4, (b and h) GOSAT, (e) OCO-2, and (c, f, and i) fused results on sodankyla01. Black dotted and red full lines stand for the 1:1 and fitted lines, respectively. X: TCCON data; Y: CAMS-EGG4/GOSAT/OCO-2/fused data. Unit: ppm/ppb to $XCO_2$/$XCH_4$ for RMSE, $\mu$, and $\sigma$.

[Figure]

**Figure 6.** Scatter-plots of the time series for daily CAMS-EGG4, GOSAT, OCO-2, fused, and TCCON data on garmisch01. The first and second numbers in the bracket represent $\mu$ and $\sigma$, respectively. Unit: ppm/ppb to $XCO_2$/$XCH_4$ for $\mu$ and $\sigma$.

[Figure]

**Figure 7.** Scatter-plots of the time series for daily CAMS-EGG4, GOSAT, OCO-2, fused, and TCCON data on lauder02. The first and second numbers in the bracket represent $\mu$ and $\sigma$, respectively. Unit: ppm/ppb to $XCO_2$/$XCH_4$ for $\mu$ and $\sigma$.

[Figure]

**Figure 8.** Heat maps of the biases between daily (a) CAMS-EGG4/(b) fused/(c) GOSAT and TCCON $XCO_2$ over time and latitude. Color ramps stand for the biases of $XCO_2$. Background colors (grey) indicate the missing data.

[Figure]

**Figure 9.** Heat maps of the biases between daily (a) CAMS-EGG4/(b) fused/(c) OCO-2 and TCCON XCO$_2$ over time and latitude. Color ramps stand for the biases of XCO$_2$. Background colors (grey) indicate the missing data.

[Figure]

**Figure 10.** Heat maps of the biases between daily (a) CAMS-EGG4/(b) fused/(c) GOSAT and TCCON $XCH_4$ over time and latitude. Color ramps stand for the biases of $XCO_2$. Background colors (grey) indicate the missing data.

[Figure]

**Figure 11.** Annual (a and g) GOSAT, (d) OCO-2, (b, e, and h) CAMS-EGG4, and (c, f, and i) fused $XCO_2$/$XCH_4$ over the globe. Color ramps stand for the values of $XCO_2$ and $XCH_4$.

[Figure]

**Figure 14.** Multi-year mean fused (a) $XCO_2$ and (b) $XCH_4$ from 2010 to 2020 over the globe. Color ramps stand for the values of $XCO_2$ and $XCH_4$.

[Figure]

**Figure 15.** Seasonal fused XCO₂ from 2010 to 2020 over the globe. The color ramp stands for the value of XCO₂. (a) DJF, (b) MAM, (c) JJA, and (d) SON denote Dec. to Feb., Mar. to May., Jun. to Aug., and Sep. to Nov., respectively.

[Figure]

**Figure 16.** Seasonal fused XCH₄ from 2010 to 2020 over the globe. The color ramp stands for the value of XCH₄. (a) DJF, (b) MAM, (c) JJA, and (d) SON denote Dec. to Feb., Mar. to May., Jun. to Aug., and Sep. to Nov., respectively.

[Figure]

**Figure 17.** Annual fused (a-k) XCO$_2$ and (l) its trend from 2010 to 2020 over the globe. Color ramps stand for the values of XCO$_2$ and its trend. ppm/yr: ppm per year.

[Figure]

**Figure 18.** Annual fused (a-k) XCH₄ and (l) its trend from 2010 to 2020 over the globe. Color ramps stand for the values of XCH₄ and its trend. ppb/yr: ppb per year.

**Q1.11:** **Text: Figure and Fig. need to be unified.**

**Response:** Thank the referee for his/her comment. In our study, the text formats of figures (i.e., "Figure" and "Fig.") followed the submission instructions of ESSD (https://www.earth-system-science-data.net/submission.html#figurestables):

- The abbreviation "Fig." should be used when it appears in running text and should be followed by a number unless it comes at the beginning of a sentence, e.g.: "The results are depicted in Fig. 5. Figure 9 reveals that...".

**Q1.12:** **References: Modify the subscript of all references.**

**Response:** Thank the referee for his/her comment. The subscript of all references has been modified.

**The main revision is as follows:**

[revised manuscript text omitted]

**Q1.13: L529: 108 000 European cities.**

**Response:** Thank the referee for his/her comment. The unreadable text of this reference has been revised in the manuscript.

**The main revision is as follows:**

Moran, D., Pichler, P.-P., Zheng, H., Muri, H., Klenner, J., Kramel, D., Többen, J., Weisz, H., Wiedmann, T., Wyckmans, A., Strømman, A. H., and Gurney, K. R.: Estimating $CO_2$ emissions for 10000 European cities, Earth System Science Data, 14, 845–864, https://doi.org/10.5194/essd-14-845-2022, 2022.

Last but not least, we gratefully thank the referee again for his/her significant comments and suggestions, which have greatly helped us to improve the technical quality and presentation of our manuscript.

---

## Author Comment (AC2)

**Response to Comments on the Manuscript (essd-2023-28):**

**Seamless mapping of long-term (2010-2020) daily global XCO$_2$ and XCH$_4$ from GOSAT, OCO-2, and CAMS-EGG4 with a spatiotemporally self-supervised fusion method**
* * *
Dear Editors and Referees,

We would like to sincerely express our gratitude to you for your careful reading and constructive comments.

According to the comments, we have tried our best to improve the manuscript, and an item-by-item response follows. The modified parts have been highlighted in yellow color in the revised manuscript.

Once again, we are particularly grateful for your careful reading and constructive comments. Thanks very much for your time.

Best regards,

Qiangqiang Yuan

**Response to Comments of Referee #2:**

**General comments:**

This study describes an effective approach to generate global long-term seamless XCO$_2$ and XCH$_4$ based on a self-supervised fusing method from OCO-2, GOSAT, and CAMS-EGG4. Generally, this paper is well organized and written, of which the methodology, validation techniques, and experiment results are reasonable. However, some details are unclear and several issues are required to be modified before this paper being published in ESSD. A major revision is recommended. Specific comments are listed as follows.

**Response:** We sincerely appreciate the referee for his/her comments and suggestions for improving the paper. An item-by-item response to the valuable comments raised by the referee follows. Thanks for your time.

**Major comments:**

**Q2.1: P9L202, Eq. (6): Is it possible to visualize intermediate results of STDCT? The visualization of intermediate results of STDCT can help understand this procedure.**

**Response:** Thank the referee for his/her comment. An example to visualize the intermediate results of *STDCT* (fusion with OCO-2 XCO$_2$) at the 20th iteration in 2017 has been presented (see Figure r1).

[Figure]

**Figure r1.** Visualization of intermediate results of *STDCT* (fusion with OCO-2 XCO$_2$) at the 20th iteration in 2017.

**Q2.2: P9L205, Eq. (7): I notice that the power of the subitem in the denominator is 1, which is different from that in the given references, such as Garcia (2010).**

**Response:** Thank the referee for his/her careful comment. The exponent (Exp) of the subitem in the denominator from Eq. (7) can be set as 1 or 2 (default) (Garcia, 2010), which controls the effect of smoothing. However, as shown in the example (see Figure r2), the fused results (such as with OCO-2 $XCO_2$) could be over-smoothed in high-latitude regions ($> 60°N$ or S) when the Exp is configured as 2. By contrast, setting this parameter as 1 will largely reduce the over-smoothing effect. Therefore, the Exp is considered 1 in our study to provide more reasonable results.

[Figure]

**Figure 2.** Daily fused $XCO_2$ using the exponents of (upper) 1 and (lower) 2 over the globe. Color ramp stands for the values of $XCO_2$.

**Reference:**

Garcia, D.: Robust smoothing of gridded data in one and higher dimensions with missing values, Computational Statistics & Data Analysis, 54, 1167–1178, https://doi.org/10.1016/j.csda.2009.09.020, 2010.

**Q2.3: Please present the whole formula derivation processes from Eq. (5) to Eq. (6).**

**Response:** Thank the referee for his/her comment. Since *STDCT* is based on a composition of one-dimensional DCTs along each dimension (Strang, 1999), the solution for one-dimensional DCT is given as an instruction as follows:

$$E(\hat{\delta}) = \left\|(\hat{\delta} - \delta)\right\|^2 + \varepsilon R(\hat{\delta}) \quad \text{(r1)}$$

where $\| \ \|$ signifies the Euclidean norm; $\delta$ and $\hat{\delta}$ are varying vectors; $\varepsilon$ indicates a smoothing factor. A simple and straightforward approach to express the roughness ($R$) is by using a second-order divided difference (Weinert, 2007, Whittaker, 1923) which yields, for a one-dimensional data array:

$$R(\hat{\delta}) = \left\|M\hat{\delta})\right\|^2 \quad \text{(r2)}$$

where $M$ is a tridiagonal square matrix defined by:

$$M_{i,i-1} = \frac{2}{s_{i-1}(s_{i-1} + s_i)}$$

$$M_{i,i} = \frac{-2}{s_{i-1}s_i}$$

$$M_{i-1,i} = \frac{2}{s_i(s_{i-1} + s_i)}$$

for $2 <= i <= n\text{-}1$, where $n$ is the number of values in $\hat{\delta}$, and $s_i$ represents the step between $\hat{\delta}_i$ and $\hat{\delta}_{i+1}$. Assuming repeating border elements ($\delta_0 = \delta_1$ and $\delta_{n+1} = \delta_n$) gives:

$$-M_{1,1} = M_{1,2} = \frac{1}{s_1^2}$$

$$-M_{n,n-1} = -M_{n,n} = \frac{1}{s_{n-1}^2}$$

At present, combining Eq. (r1) and (r2) acquires:

$$(I_n + \varepsilon M^T M)\hat{\delta} = \delta \quad \text{(r3)}$$

where $I_n$ is the $n \times n$ identity matrix and $M^T$ stands for the transpose of $M$. Eq. (r3) can be further extended to multi-dimensional regularly gridded data using DCTs, which is much more complicated. More detailed descriptions for the extension of Eq. (r3) are provided in Buckley (1994) and Garcia (2010).

**Reference:**

Buckley, M.J.: Fast computation of a discretized thin-plate smoothing spline for image data, Biometrika, 81, 247–258, https://doi.org/10.1093/biomet/81.2.247, 1994.

Garcia, D.: Robust smoothing of gridded data in one and higher dimensions with missing values, Computational Statistics & Data Analysis, 54, 1167–1178, https://doi.org/10.1016/j.csda.2009.09.020, 2010.

Strang, G.: The discrete cosine transform, SIAM Review, 41, 135–147, https://www.jstor.org/stable/2653173, 1999.

Weinert, H.L.: Efficient computation for Whittaker–Henderson smoothing, Computational Statistics & Data Analysis, 52, 959–974, https://doi.org/10.1016/j.csda.2006.11.038, 2007.

Whittaker, E.T.: On a new method of graduation, Proceedings of the Edinburgh Mathematical Society, 41, 62–75, https://doi.org/10.1017/S0013091500077853, 1923.

**Q2.4: P9L209: Different initializations of $\widehat{\delta}$ may lead to different final results. Please provide a brief discussion?**

**Response:** Thank the referee for his/her constructive comment. At first, we would like to apologize for that the description of the initializations for missing values in $\widehat{\delta}$ is misleading in our study, which has been revised. Due to the large sparsity of satellite $XCO_2$ and $XCH_4$, different initializations of $\widehat{\delta}$ will cause non-negligible differences in fused results. Here, an example of three initializations is presented to tell the differences among them, defined as follows:

1. Type 1: spatiotemporal nearest neighbor interpolation (adopted in our study).

2. Type 2: temporal nearest neighbor interpolation.

3. Type 3: replacement with a constant value (e.g., 1).

[Figure]

**Figure r3.** Schematic diagram to show the iteration process of $\delta$ (fusion with OCO-2 $XCO_2$) via Type 1 initialization in 2017-04-10.

[Figure]

**Figure r4.** Schematic diagram to show the iteration process of $\delta$ (fusion with OCO-2 XCO$_2$) via Type 2 initialization in 2017-04-10.

[Figure]

**Figure r5.** Schematic diagram to show the iteration process of $\delta$ (fusion with OCO-2 XCO$_2$) via Type 3 initialization in 2017-04-10.

Figure r3-r5 demonstrate the iteration processes of $\delta$ (fusion with OCO-2 XCO$_2$) using three types of initializations in 2017-04-10. It is clear that different initializations of $\hat{\delta}$ could generate similar but different results. Figure r6 illustrates the in-situ validation results of the fused results with OCO-2 XCO$_2$ during 2015-2020 through three types of initializations. As observed, the fused XCO$_2$ using Type 1 initialization achieves the best performance, which signifies that more prior information in the initialization can improve the fusion accuracy.

[Figure]

**Figure r6.** Density scatter-plots of the in-situ validation results for fused $XCO_2$ with OCO-2 using (left) Type 1, (middle) Type 2, and (right) Type 3 initialization from 2015 to 2020. Black dotted and red full lines stand for the 1:1 and fitted lines, respectively. Color ramps show the normalized densities of data points. X: TCCON data; Y: fused data. Unit: ppm for RMSE, $\mu$, and $\sigma$.

**The main revision is as follows:**

It is worth noting that $\hat{\delta}$ is initialized through the spatiotemporal nearest neighbor interpolation.

**Q2.5: Will the data completeness of XCO₂/XCH₄ from OCO-2/GOSAT affect the accuracy of final fused results? More data should imply more usable information. Please provide a brief discussion.**

**Response:** Thank the referee for his/her crucial comment. An example to show the in-situ validation results of the fused results with OCO-2 $XCO_2$ (different data completeness) during 2015-2020 is depicted in Figure r7. As can be seen, the fusion with OCO-2 $XCO_2$ of less data completeness (20-80% discarded) can variously reduce the performance. However, the fused results still present better accuracy than that of CAMS-EGG4, which indicates the robustness of the proposed fusion method.

[Figure]

**Figure r7.** Density scatter-plots of the in-situ validation results for CAMS-EGG4 and fused $XCO_2$ with OCO-2 (different data completeness) from 2015 to 2020. Black dotted and red full lines stand for the 1:1 and fitted lines, respectively. Color ramps show the normalized densities of data points. X: TCCON data; Y: CAMS-EGG4/fused data. Unit: ppm for RMSE, $\mu$, and $\sigma$.

**Q2.6:** **The data of $XCO_2$/$XCH_4$ from OCO-2/GOSAT is extremely sparse in some regions, I wonder if the performance could be improved after fusion under this condition.**

**Response:** Thank the referee for his/her crucial comment. Same to **Q2.5** (see 80% discarded in Figure r7), the fused results with extremely sparse data present a decreased accuracy, which is still superior to that of CAMS-EGG4.

**Q2.7:** **Table 3-5: The metrics of the individual in-situ validation do not exceed those of CAMS-EGG4 for a few stations after fusion. What could be the potential reasons? Please provide a further discussion.**

**Response:** Thank the referee for his/her comment. The performance will reduce for a few stations after fusion, which is mainly affected by the poor quality of satellite $XCO_2$ and $XCH_4$. For instance (see Figure r8), all the metrics (e.g., $R^2$, RMSE, and $\sigma$) of fused $XCO_2$ with GOSAT are worse than those

of CAMS-EGG4 on tsukuba02 from 2010 to 2014. This is likely attributed to the generally underestimated values of GOSAT $XCO_2$ (i.e., $\mu$: -1.094 ppm).

[Figure]

**Figure r8.** Scatter-plots of the in-situ validation results for CAMS-EGG4 (left), GOSAT (middle), and fused $XCO_2$ (right) on tsukuba02 during 2010-2014. Black dotted and red full lines stand for the 1:1 and fitted lines, respectively. X: TCCON data; Y: CAMS-EGG4/GOSAT/fused data. Unit: ppm for RMSE, $\mu$, and $\sigma$.

**Minor comments:**

**Q2.8: P5L121: The authors did not consider the latest $XCO_2$ from OCO-3 for fusion. What is the reason?**

**Response:** Thank the referee for his/her comment. The latest $XCO_2$ from OCO-3 presents a similar accuracy to that from OCO-2 with a shorter period (available after August 2019) (Taylor et al., 2023). As a result, the OCO-2 $XCO_2$ product is currently employed for fusion in this study. The $XCO_2$ from OCO-3 can be considered in our future works.

**Reference:**

Taylor, T. E., O'Dell, C. W., Baker, D., Bruegge, C., Chang, A., Chapsky, L., Chatterjee, A., Cheng, C., Chevallier, F., Crisp, D., Dang, L., Drouin, B., Eldering, A., Feng, L., Fisher, B., Fu, D., Gunson, M., Haemmerle, V., Keller, G. R., Kiel, M., Kuai, L., Kurosu, T., Lambert, A., Laughner, J., Lee, R., Liu, J., Mandrake, L., Marchetti, Y., McGarragh, G., Merrelli, A., Nelson, R. R., Osterman, G., Oyafuso, F., Palmer, P. I., Payne, V. H., Rosenberg, R., Somkuti, P., Spiers, G., To, C., Wennberg, P. O., Yu, S., and Zong, J.: Evaluating the consistency between OCO-2 and OCO-3 $XCO_2$ estimates derived from the NASA ACOS version 10 retrieval algorithm, AMTD, https://doi.org/10.5194/amt-2022-329, 2023.

**Q2.9: P5L130: Similarly, the authors also did not adopt the popular $XCO_2$ from Carbon Tracker**

**for fusion.**

**Response:** Thank the referee for his/her comment. The $XCO_2$ from the Carbon Tracker (https://gml.noaa.gov/ccgg/carbontracker/) is popular but performed at a coarse spatial resolution of $3° \times 2°$. In addition, the Carbon Tracker only provides $XCH_4$ from 2000 to 2010. By contrast, the CAMS-EGG4 $XCO_2$ and $XCH_4$ products are more appropriate and adopted in this study. The $XCO_2$ and $XCH_4$ from the Carbon Tracker can be considered in our future works.

**Q2.10:** **The figures in the Supplement are too many to follow, which are unnecessary. Table 3-5 have summarized their metrics.**

**Response:** Thank the referee for his/her comment. The figures in the Supplement have been removed from the manuscript.

**Q2.11:** **Table 3-5: It's better to abbreviate "CAMS-EGG4" into "CAMS" instead of "CE", which is consistent with other texts.**

**Response:** Thank the referee for his/her careful comment. "CAMS-EGG4" has been abbreviated into "CAMS" in Table 3-5.

**The main revision is as follows:**

**Table 3.** Metrics of the individual in-situ validation results for CAMS-EGG4, GOSAT, and fused $XCO_2$. The best and second metrics are denoted with bold and underlined fonts. CAMS: CAMS-EGG4; AF: after fusion. Unit: ppm for RMSE and $\sigma$.

**Table 4.** Metrics of the individual in-situ validation results for CAMS-EGG4, OCO-2, and fused $XCO_2$. The best and second metrics are denoted with bold and underlined fonts. CAMS: CAMS-EGG4; AF: after fusion. Unit: ppm for RMSE and $\sigma$.

**Table 5.** Metrics of the individual in-situ validation results for CAMS-EGG4, GOSAT, and fused $XCH_4$. The best and second metrics are denoted with bold and underlined fonts. CAMS: CAMS-EGG4; AF: after fusion. Unit: ppb for RMSE and $\sigma$.

**Q2.12:** **Future works and limitations are missing in the Conclusions.**

**Response:** Thank the referee for his/her comment. Future works and limitations have been appended in the manuscript.

**The main revision is as follows:**

Overall, the developed fusion method generates high-quality full-coverage $XCO_2$ and $XCH_4$ datasets over the globe from 2010 to 2020. However, it only considers the global spatiotemporal knowledge of self-correlation in GOSAT and OCO-2 products without attention to local spatiotemporal information. Meanwhile, the spatial resolution and available period of fused results should be further enhanced, which are devised as 0.1° and more than 20 years (e.g., 2000-2020), respectively. To fix these issues, we will spare no effort to work on our future works.

**Q2.13:** **Is it feasible to acquire global seamless $XCO_2$ and $XCH_4$ only from OCO-2 and GOSAT based on the proposed method?**

**Response:** Thank the referee for his/her comment. Generating global seamless $XCO_2$ and $XCH_4$ only from OCO-2 and GOSAT is still a challenge due to their significant sparsity without any external data. At present, the proposed method merely can provide some over-smoothed results, which are required to be improved in our future works.

Last but not least, we gratefully thank the referee again for his/her significant comments and suggestions, which have greatly helped us to improve the technical quality and presentation of our manuscript.